# Co-Amorphous Drug Formulations in Numbers: Recent Advances in Co-Amorphous Drug Formulations with Focus on Co-Formability, Molar Ratio, Preparation Methods, Physical Stability, In Vitro and In Vivo Performance, and New Formulation Strategies

**DOI:** 10.3390/pharmaceutics13030389

**Published:** 2021-03-15

**Authors:** Jingwen Liu, Holger Grohganz, Korbinian Löbmann, Thomas Rades, Nele-Johanna Hempel

**Affiliations:** Department of Pharmacy, University of Copenhagen, 2100 Copenhagen, Denmark; jingwen.liu@sund.ku.dk (J.L.); holger.grohganz@sund.ku.dk (H.G.); korbinian.loebmann@sund.ku.dk (K.L.); nele.hempel@sund.ku.dk (N.-J.H.)

**Keywords:** co-amorphous, poorly water soluble drugs, solid dispersion, co-formability, in vivo studies, physical stability, molar ratio optimization

## Abstract

Co-amorphous drug delivery systems (CAMS) are characterized by the combination of two or more (initially crystalline) low molecular weight components that form a homogeneous single-phase amorphous system. Over the past decades, CAMS have been widely investigated as a promising approach to address the challenge of low water solubility of many active pharmaceutical ingredients. Most of the studies on CAMS were performed on a case-by-case basis, and only a few systematic studies are available. A quantitative analysis of the literature on CAMS under certain aspects highlights not only which aspects have been of great interest, but also which future developments are necessary to expand this research field. This review provides a comprehensive updated overview on the current published work on CAMS using a quantitative approach, focusing on three critical quality attributes of CAMS, i.e., co-formability, physical stability, and dissolution performance. Specifically, co-formability, molar ratio of drug and co-former, preparation methods, physical stability, and in vitro and in vivo performance were covered. For each aspect, a quantitative assessment on the current status was performed, allowing both recent advances and remaining research gaps to be identified. Furthermore, novel research aspects such as the design of ternary CAMS are discussed.

## 1. Introduction

The majority of drug candidates in pharmaceutical development exhibit poor water solubility and are categorized as Class II or Class IV drugs based on the Biopharmaceutics Classification System (BCS) [1,2]. BCS Class II drugs show poor solubility, whilst BCS Class IV drugs additionally show poor permeability. One promising approach to overcome the poor solubility of drugs is the transformation of the drug from a crystalline state to an amorphous form [3]. The amorphous form is characterized by enhanced dissolution properties, namely a higher apparent solubility as well as a higher dissolution rate. On the downside, the amorphous form is thermodynamically unstable; i.e., the amorphous form of a drug will recrystallize into a crystalline state over time, making it necessary for the amorphous form to be stabilized to exploit the enhanced dissolution properties [4,5]. A common approach for the stabilization of amorphous forms of a drug is the formation of a solid dispersion [6,7].

Solid dispersions were first described in 1971 by Chiou and Riegelman [8] and are defined via the stabilization of the amorphous form of a drug by an inert carrier in the solid state. Solid dispersions are divided into several subcategories, wherein amorphous solid dispersions (ASDs) form the majority of cases investigated in the pharmaceutical literature. An ASD is defined as a solid dispersion that is amorphous after preparation. Commonly, in the literature, polymer-based ASDs are used synonymously with the expression ASD; however, mesoporous silica-based glass solutions and co-amorphous systems are included in the concept of ASDs as well [9].

Polymer-based ASDs are dominant in the field of ASDs, and extensive research over the past 50 years has resulted in several marketed products [2,10]. However, despite substantial research efforts, polymer-based ASDs have several shortcomings, such as low drug-loadings of approximately 20–30 wt %, which limit their suitability to highly potent drugs [7,11]. Furthermore, some polymers, such as polyvinylpyrrolidone (PVP), are hygroscopic, leading to water sorption during storage causing phase-separation and recrystallization [7].

Research interest in mesoporous silica-based ASDs is increasing; however, no pharmaceutical product has made it to the market yet [12]. Only recently, research has started to investigate the stabilization mechanism of drugs in mesoporous silica-based ASD [13]. Despite the possibility of achieving a higher drug load than in polymer-based ASD, it has been shown that only the monomolecular drug layer with direct contact to the silanol groups inside the mesopores of mesoporous silica-based ASD is fully stabilized [13,14,15]. Additional drug loadings inside the mesopores can recrystallize depending on the pore diameter or move out of the pores to recrystallize [16,17]. Further research is necessary to fully elucidate the potential of mesoporous silica-based ASD [12,17].

Co-amorphous ASDs, here referred to as co-amorphous systems (CAMS), are defined by the stabilization of the amorphous form of a drug by one or more low-molecular weight excipients and/or other drug molecules, the co-former(s), and their formation of a homogenous single-phase amorphous system. This approach has some potential advantages over polymer-based and mesoporous silica-based ASD as the drug load can be increased from 20–30 wt % to approximately 50 wt % or even higher in many cases (see Section 6). The low-molecular weight co-formers may be other drug substances or low molecular weight excipients such as amino acids, organic acids, and other small molecules, such as nicotinamide (see Section 3). The co-former physically stabilizes the amorphous form of the drug either by interacting with the drug on a molecular level (e.g., by salt formation, hydrogen bonding, and pi–pi interactions; see for example [18,19]) or simply by molecular mixing (see for example [18,19,20,21]).

CAMS can be described by three critical quality attributes (CQA), namely, co-formability, physical stability, and dissolution performance (Figure 1). The first CQA describes the possibility of forming a CAMS consisting of a drug and a chosen co-former at a defined ratio of drug and co-former for a defined preparation method. The second CQA describes the physical stability, as the CAMS must increase the physical stability of the otherwise thermodynamically unstable amorphous form of the pure drug. Finally, the third CQA has to be considered; i.e., the degree in dissolution enhancement must be compared to the pure drug in both the crystalline state and the amorphous form. Enhanced dissolution performance is defined by obtaining (and maintaining) supersaturation, resulting potentially in a higher bioavailability in vivo (see Section 9).

This review aims to give an overview of the developments in the field of CAMS research with a focus on these three CQA. In particular, the review focuses on co-formability, the choice of the molar ratio between drug and co-former, preparation methods, the physical stability of CAMS, and the correlation between in vitro and in vivo performance. Moreover, a future outlook is presented including recent advances in newly designed formulations to optimize the three CQA.

## 2. Research Interest in the Field of CAMS

The first CAMS were reported as binary amorphous systems and date back to before the year 2000 and are often overlooked, as these reports solely looked into thermal analysis of binary amorphous mixtures, which however, is part of the first CQA [22,23,24,25]. The term “co-amorphous” was first introduced by Chieng et al. in 2009 [26]. Shortly thereafter, the number of published studies on CAMS increased. The diagram below shows the relevant studies used for this review sorted by year starting in 1989 (Figure 2).

It can be seen from Figure 2 that the research interest in CAMS increased over the past years, leading to nearly 200 relevant studies and additionally several reviews (not included in Figure 2). An overview of the studies included in this review can be found in Appendix C. Appendix D gives an overview of relevant review articles, which were also consulted for this review. In general, it is expected that the number of published studies on CAMS will continue to increase in the future.

To determine how systematically CAMS have been investigated, the amount of screened drug and co-former combinations per study was counted. We found that nearly 50% of all studies only investigated one drug/co-former combination. In total, 83.3% of the included studies investigated four or less drug/co-former combinations. Only 16.7% of the included studies included five or more drug/co-former combinations (Note: this number includes studies conducted with a theoretical or computational modeling approach). In other words, half of the studies investigating CAMS are case studies with single examples of one possible drug/co-former combination. The lack of systematic studies to investigate the broad feasibility of this stabilization approach for the amorphous form of a drug may, at least in part, explain the lack of marketed products. The lack of systematic studies has already been mentioned in a previous review article by Korhonon et al. (2017), stating that the use if in silico tools should be exploited more in the research field of CAMS to predict suitable drug/co-former combinations [27].

## 3. Classes of Investigated CAMS

CAMS can be categorized into several (sub-)classes according to the nature of their co-formers, such as drug–drug CAMS, drug–amino acid CAMS, drug–organic acid CAMS, and drug–other excipient CAMS.

As shown in Figure 3, amino acids are currently the most used co-formers in CAMS, accounting for 46% of the reported CAMS (Appendix B). Initially, amino acids have been employed based on the knowledge of possible drug–amino acid interactions at biological binding sites of receptors [18,28]. Subsequently, it was clarified that the presence of an amino acid from the receptor binding site or the presence of strong molecular interactions (i.e., ionic interactions) was not a prerequisite for the successful formation of drug–amino acid CAMS [18,29,30]. In general, even without distinct interactions, CAMS can still contribute to the improvement of physical stability as a result of molecular mixing [9,18,19,20,21].

The fraction of reported drug–drug CAMS in the literature is 23.6%. Drug–drug CAMS have been designed based on pharmacokinetic and pharmacodynamic properties of the drugs. Apart from the advantage of dissolution and stability improvements, drug–drug CAMS provide a platform to achieve potential combination therapies. The first drug–drug CAMS were reported as binary glass systems in 1989 [22]. Following their findings, the authors dug further and investigated the formation of drug–drug CAMS by combining cimetidine with naproxen [24]. However, the application of drug–drug CAMS may be limited, as pharmacologically relevant drug–drug combinations cannot always be given at fixed doses.

Drug–organic acid CAMS form the third largest class of the reported CAMS with 12.9%. Organic acids have been applied as co-formers in CAMS mainly as acidic excipients to form strong molecular interactions with basic drugs (for example in [25,31,32,33,34,35,36,37,38,39,40,41,42,43]). The organic acids used as co-formers differ in their number of carboxylic acid groups, which was hypothesized to lead to salt formation with a basic drug at different molar ratios. The formation of carvedilol–organic acids CAMS at various molar ratios highlighted the possibility of achieving a higher drug to organic acid ratio when using di-, or triprotic organic acids instead of monoprotic organic acids [39].

Apart from the above-mentioned co-former classes, other types of small molecules have been applied in the preparation of CAMS, for example urea [41,44,45] and nicotinamide [41,43,45,46,47,48,49]. These CAMS contribute 17.5% of the total. Overviews of these screened (non-) formations of CAMS using amino acids, drugs, organic acids, and other excipients as co-formers are listed in the Appendix A.

Furthermore, it is worth mentioning that phospholipids and sugars have been employed as co-formers to improve the physical stability and/or increase the dissolution properties of drugs. Interestingly, phospholipids and sugars mainly acted as matrix in these solid dispersions, and the drug concentrations were considerably lower than in other CAMS, in some cases similar to polymer-based ASDs. In addition, most of these systems did not show clear evidence of the existence of one homogeneous amorphous phase. Therefore, these systems were not included in the CAMS dataset for this review. Instead, the studies reporting (amorphous) systems with phospholipids and sugars are listed in Appendix E.

## 4. Drugs Investigated for the Formation of CAMS

A total of 129 different drugs have been screened for their use in CAMS. Out of these 129 different compounds, nearly half of them (43.8%) have only been reported once (Figure 4a), and less than one-quarter (23.4%) have been used in more than five different studies (Figure 4a). This underlines a lack of systematic investigation and points toward the investigation of individual examples case by case (Note: every drug was only included once from Pajula et al. (2010), who were using a computational approach [50]). The top five drugs investigated in CAMS were, in decreasing order, indomethacin, mebendazole, carvedilol, carbamazepine, and simvastatin (which have been used in 49.0% of the screened drug/co-former combinations) (Note: a co-former can also be a drug here) (Figure 4b). The top five drugs all belong to BCS Class II. Whilst this shows the good applicability of CAMS for this class of drugs, it nonetheless points toward a lack of drug investigations based on generalizable drug properties. In other words, it is unclear if the behavior of those five drugs in CAMS with respect to co-formability, physical stability, and dissolution behavior can be directly translated to other drugs.

## 5. Co-Former Selection of Investigated CAMS

The selection of an appropriate co-former is crucial for the preparation of a desired CAMS, since it determines the first CQA, the co-formability (Figure 1). The criteria of co-former selection in the reported CAMS are shown in the Figure 5. The biggest proportion of the co-formers in CAMS (37.4%) were chosen based on previous studies. Considerations of bioactivity, structural characteristics of drug and co-former, and physicochemical properties of co-formers on the one hand and consideration of combination therapy on the other hand accounted for 18.1% and 6.8%, respectively. However, a systematic co-former screening was only performed in 21.8% of the reported studies [29,36,41,50,51,52,53,54,55,56,57]. Therefore, systematic, predictive, and computational screening methods for co-former selections are still largely unexplored (see also a review: [27]). The reported criteria for co-former selection in CAMS are summarized in the following section of this review.

### 5.1. Methods for Co-Former Selection

#### 5.1.1. Prediction of the Miscibility of Two Components

Miscibility between two selected components is one of the most important aspects to form stable amorphous systems. Calculations of the Hansen solubility parameter and the Flory–Huggins interaction parameter have been applied to preliminarily estimate the miscibility of drug and the selected co-formers. Two components are likely to be miscible with each other if the difference in their solubility parameters is smaller than 7.0 MPa [27,58], or their Flory–Huggins interaction parameter has a negative or only slightly positive value [27,59,60].

Solubility parameters were used as a tool to predict the solubility of ibuprofen and ibuprofen lysinate in small sugar molecule carriers [61]. Solubility parameters were determined by using both calculation-based methods and experimental approaches, and the obtained values were similar between these two. The results predicted that ibuprofen would be insoluble in the sugar excipients, while ibuprofen lysinate would be soluble. Unfortunately, these predictions have not been verified by experiments. In addition, a data mining tool was developed to predict the miscibility of indomethacin and several excipients (more than 30 compounds) based on their Hansen solubility parameters, and thereafter, the accuracy of this method was evaluated experimentally [62]. Physical mixtures of 2:1, 1:1, and 1:2 molar ratios of drug and excipients were tested by thermal analysis (differential scanning calorimetry, DSC), and the formation of a eutectic mixture upon heating was taken as an indicator for miscibility. It was found that the prediction correlated well with the experimental results, with 94% accuracy.

The Flory–Huggins interaction parameter was used to evaluate the miscibility between simvastatin and glipizide [20]. However, the successful formation of a CAMS was achieved even though the calculated Flory–Huggins interaction parameter was positive, indicating that these two drugs would be expected to be immiscible. Therefore, the increased physical stability of this system, compared to pure amorphous drugs and the amorphous physical mixtures, could not be explained by thermodynamic miscibility.

On a large scale, the calculation of the Flory–Huggins interaction parameters has been used as a screening method for CAMS. Pajula et al. (2010) computationally determined the Flory–Huggins interaction parameters and phase diagrams of 1122 CAMS, and it was demonstrated that the Flory–Huggins interaction parameters could be regarded as a relatively good indicator for fast screening on a large drug and co-former set [50]. Taking conformational variations of the molecules into consideration, the authors further modified the in silico method for predicting miscibility based on the Flory–Huggins interaction parameters [53]. They used this method to predict the immiscibility of the binary mixtures and thereafter detected amorphous–amorphous phase separation experimentally by a Fourier transform infrared spectroscopy (FTIR) imaging technique. This method was further applied to design model mixtures for amorphous–amorphous phase separation investigation of CAMS; the authors designed four expected immiscible pairs based on the Flory–Huggins methods and thereafter managed to detect early stage amorphous–amorphous phase separation in CAMS using a scanning electron microscope with energy dispersive X-ray spectrometer (SEM-EDS) [55]. Investigation on amorphous–amorphous phase separation is important for the detection of physical (in-)stability at the early storage stage, which has been widely studied in ASDs [63,64,65,66]. However, only a few studies have been conducted on this phenomenon in CAMS [53,55,67,68]. This or similar methods should be used further to expand the research on the phase separation of CAMS in the future.

#### 5.1.2. Physicochemical Properties of Co-Formers

The contribution of some physicochemical properties of the co-formers to drug–drug CAMS formation was evaluated by using multivariate (partial least squares discriminant (PLS-DA)) analysis [51]. The work emphasized the importance of glass forming ability, crystallization tendency, glass transition temperature (*T*_g_), melting point, molecular flexibility, hydrogen bonding acceptor number, and topological polar surface area of the co-formers for this class of CAMS.

It was also investigated which molecular descriptors are of importance to predict the likelihood of successful co-amorphization between drugs and amino acids as co-formers [54]. The descriptor differences for the chosen drug–amino acid mixtures (based on [29]) were calculated and used as input for a PLS-DA. The model showed a clear separation of the co-amorphous and the not co-amorphous samples and underlined that amino acids with non-polar side chains were dominating the co-amorphous system class. Recently, a further step was taken to optimize and expand the application of this method by reducing the number of variables from 36 to 7 and testing other co-formers (rather than only amino acids) [41]. The method showed potential to predict the co-formability of different co-formers with a high accuracy.

In addition, a model to predict the glass forming ability and crystallization tendency of 131 selected compounds was developed based on the physicochemical properties of the components, such as molecular weight, the number of hydrogen bond acceptors, the Huckel π atomic charges, and many more [69]. The models found that 77 compounds out of the 131 compounds formed solid dispersions, which could be used to choose poorly soluble drug molecules with glass forming ability, especially for drug–drug CAMS.

In addition to the methods above, it is worth mentioning that pK_a_ values of the co-formers play a critical role in the amorphous salt formation [29]. A difference of >2 or 3 in pK_a_ values between an acid and a base can result in a salt formation [70]. Therefore, the pK_a_ values should be considered at the co-former selection stage to understand whether amorphous salts can be formed.

### 5.2. Stabilization Mechanism of CAMS

The stabilization mechanism of CAMS varies based on the different selections of co-formers. As mentioned above, the stabilization mechanism can be via (amorphous) salt formation, when combining an acidic drug with a basic co-former or vice versa. Nevertheless, also weaker molecular interactions can contribute to the stabilization in a CAMS, such as hydrogen bonding and π–π interactions. Interestingly, the existence of interactions is not a prerequisite for the formation of a CAMS. Several cases have been described, where no molecular interactions between the drug and the co-former could be found. CAMS not showing signs of molecular interactions between the drug and the co-former can still stabilize the amorphous drug by physical separation of similar molecules, referred to as molecular or intimate mixing [9,20,21]. In addition, good miscibility between two components in the solid state and elevated glass transition temperatures (*T*_g_s) of CAMS compared to their individual amorphous components have also been regarded as contributors for improved physical stability. Stabilization mechanisms of CAMS have been discussed in detail in previous reviews [9,71], and interested readers are referred to these reviews.

## 6. Molar ratio Optimization of Investigated CAMS

After the initial co-former selection, stoichiometry plays an important role in the preparation of CAMS, since this further influences the three CQA. As shown in Figure 6, the molar ratio 1:1 is the most commonly used stoichiometric ratio in CAMS, with more than 70% of the reported CAMS prepared at equimolar ratio. Only 23.1% of the CAMS were prepared at various molar ratios. After a further analysis of the CAMS with different molar ratios, it was found that a molar ratio optimization process (with respect to the second or/and third CQA of CAMS) was applied only in 5.8% of the studies, while the remaining 17.3% chose several specific molar ratios (such as the molar ratios 1:2, 1:1, and 2:1) without further justification.

A summary of the studies reporting CAMS, in which a molar ratio optimization was applied, is shown in Table 1. In approximately half of the cases, where the molar ratio was optimized, it was shown that the ideal molar ratio (with respect to physical stability or/and dissolution performance of CAMS) between drug and co-former was not at the equimolar ratio. For example, for the carvedilol and aspartic acid CAMS, it was shown that the 1:1 molar ratio was not the ideal molar ratio for physical stability, even though salt formation between carvedilol and aspartic acid was expected to occur at the equimolar ratio based on the chemical nature of the drug and co-former [72].

Furthermore, it can be seen that often a drug load above 50 wt % can be achieved. For example, carvedilol–organic acid CAMS have their optimal molar ratio at a drug loading of 85.9 wt % with malic acid, 83.3 wt % with benzoic acid, and 80.83 wt % with citric acid [39]. In drug–amino acid CAMS, the drug load can be above 50 wt %, as for example in furosemide–arginine, even at the equimolar ratio of 1:1, which corresponds to a drug loading of 65.5 wt % [30]. For the drug–drug CAMS naproxen–indomethacin at the optimal molar ratio of 1.5:1, the drug load of naproxen is at 49.1 wt % [73,74]. In general, using low weight excipients, such as organic acid or amino acid, the drug load can be increased to ≥ 50 wt %. Compared to polymer-based ASD, the drug loading can be highly improved using CAMS, which can be beneficial for downstream manufacturing as lower amounts of excipients are necessary.

Based on the results listed in Table 1, the importance of molar ratio optimization in preparation of CAMS is highlighted. At an optimized ratio, it is suggested that all of the three CQA can be positively affected. However, this requires further investigation.

### 6.1. Methods for Molar Ratio Optimization

#### 6.1.1. Detection of Endothermic and Exothermic Thermal Events

Molar ratio optimization of CAMS can be conducted based on the thermal behavior of the physical mixtures of the CAMS. Using thermal analysis (DSC) of binary physical mixtures of the CAMS at various molar ratios, the melting endotherms and/or recrystallization exotherms of the drug and co-former can be detected. The absence of melting endotherms or/and recrystallization exotherms of both drug and co-former was regarded as an indicator of no excess amount of drug and co-former in the physical mixture of the CAMS at the specific molar ratio and hence, it was defined as optimal molar ratios [75,76,77]. This thermal analysis screening is simple; however, depending on the choice of the different molar ratios of the physical mixtures, the optimal molar ratio range can be broad.

**Table 1 pharmaceutics-13-00389-t001:** Overview of studies using molar ratio optimization to find the optimal molar ratio for the investigated amorphous systems. Y: Yes, N: No. *ROY: 5-methyl-2 [(2-nitrophenyl)-amino]-3-thiophenecarbonitrile. DSC: Differential scanning calorimetry. DMA: Dynamical mechanical analysis. PCA: Principal component analysis. XRPD: X-ray powder diffractometry. FTIR: Fourier-transform infrared spectroscopy. *T*_g_: glass transition temperature.

Amorphous Systems	Preparation Method	Compositions of the Systems	Optimization Methods	The Optimal Molar Ratio(s)	Physical Stability to Confirm the Optimal Molar Ratio	Reference
Atenolol- Urea	Melt-quench	Molar ratios of 1:1, 1:2, 1:4, 1:6, 1:8, 1:10, 1:12	Thermal analysis by DSC; Precipitation test	Atenolol–Urea 1:4 for CAMS; Atenolol–Urea 1:8 for super-saturation maintenance	N	[44]
Carvedilol– Aspartic acid	Spray drying	Molar ratios of 2:1, 1:1, 1:1.25, 1:1.5, 1:1.75, 1:2, 1:2.25, 1:2.5, 1:3, 1:4	Data fitting methods of *T*_g_s; FTIR-PCA	1:1.46 (mathematically); 1:1.5 (experimentally)	Y	[72]
Carvedilol– Benzoic acid	Spray drying	Molar ratios of 1:4, 1:3, 1:2, 1:1, 1.5:1, 2:1, 4:1	Determination of the highest *T*_g_	1.5:1	Y	[39]
Carvedilol– Citric acid	Spray drying	Molar ratios of 1:4, 1:3, 1:2, 1:1, 2:1, 3:1, 4:1	Determination of the highest *T*_g_	2:1	Y	[39]
Carvedilol– Glutamic acid	Spray drying	Molar ratios of 2:1, 1:1, 1:1.25, 1:1.5, 1:1.75, 1:2, 1:2.25, 1:2.5, 1:3, 1:4	Data fitting methods of *T*_g_s; FTIR-PCA	1:1.43 (mathematically); 1:1.5 (experimentally)	Y	[72]
Carvedilol– Malic acid	Spray drying	Molar ratios of 1:4, 1:3, 1:2, 1:1, 2:1, 3:1, 4:1	Determination of the highest *T*_g_	2:1	Y	[39]
Carvedilol– Tryptophan	Ball milling	Molar fractions 0.1–0.9, at an interval of 0.1	Determintion of *T*_g_*β* by DMA; thermal analysis by DSC (lack of any endothermic or exothermic events)	The molar fractions of carvedilol were 34–52% (equal to the molar ratio of Carvedilol–Tryptophan from 1:0.92 to 1:1.94).	Y	[73]
Ezetimibe– Lovastatin– Soluplus^®^	Spray drying	Ezetimibe–Lovastatin at the weight ratios of 1:1, 1:2, 1:4. The weight fractions of soluplus^®^ were 50 wt %, 75 wt %, 90 wt %.	Physical stability	Weight ratio of 12.5:12.5:75.	Y	[78]
Ezetimide–Simvastatin– Kollidon^®^ VA64	Melt-quench	Ezetimibe–Simvastatin at the weight ratios of 1:1. The weight fractions of polymer were 5 wt %, 20 wt %, 40 wt %, 60 wt %	The viscoelastic properties measured by oscillatory shear rheology	Minimal 40 wt % polymer required	N (only confirmed CAMS with 40 wt % polymer was stable)	[79]
Furosemide– Arginine	Ball milling	Molar fractions of furosemide from 0.09 to 0.9	Comparison of the experimental *T*_g_s to the theoretical *T*_g_s for the largest deviation	1:1	N	[30]
Furosemide– Tryptophan	Ball milling	Molar fractions of furosemide from 0.09 to 0.9	Comparison of the experimental *T*_g_s to the theoretical *T*_g_s for the largest deviation	1:1	N	[30]
Indomethacin– Arginine	Ball milling	Molar fractions of indomethacin from 0.09 to 0.9	Comparison of the experimental *T*_g_s to the theoretical *T*_g_s for the largest deviation	1:1	N	[30]
Indomethacin–Naproxen	Melt-quench	Molar fractions 0.1–0.9, at an interval of 0.1	Phase diagrams to determine the eutectic point	1:1.5	Y	[73]
Indomethacin–Tryptophan	Ball milling	Molar fractions of indomethacin from 0.09 to 0.9	Comparison of the experimental *T*_g_s to the theoretical *T*_g_s for the largest deviation	1:1	N	[30]
Indomethacin–Tryptophan	Ball milling	Molar fractions 0.1–0.9, at an interval of 0.1	Determintion of *T*_g_*_β_* by DMA; thermal analysis by DSC (lack of any endothermic or exothermic events)	The molar fractions of indomethacin were 5–25% (equal to the molar ratio of Indomethacin–Tryptophan from 1:3 to 1:19).	Y	[76]
Naproxen– Indomethacin	Melt-quench	Molar fractions 0.1–0.9, at an interval of 0.1	XRP–diffractograms–PCA; FTIR–PCA; Phase diagrams	1.5:1	Y	[74]
Naproxen– Meglumine	Melt-quench	Molar ratios of 10:1, 2.5:1, 10:7, 1:1, 7:10, 1:2.5, 1:10	Determination of the highest glass transition temperature; Physical stability	1:1	Y	[80]
Naproxen–Sodium– Indomethacin	Melt-quench	Molar fractions 0.1–0.9, at an interval of 0.1	Physical stability	Naproxen–Sodium: 0.1–0.4 (equal to the molar ratio Naproxen–Sodium:Indomethacin from 1:9 to 1:1.5)	Y	[81]
Naproxen–Sodium– Naproxen–Indomethacin	Melt-quench	Molar ratio of Naproxen–Sodium:Naproxen fixed at 1:1; Molar fractions of Indomethacin: 0.1–0.9, at an interval of 0.1	Physical stability	Indomethacin: 0.3–0.9 (equal to the molar ratio Naproxen–Sodium:Naproxen:Indomethacin from 1:1:0.86 to 1:1:1.8)	Y	[81]
Nifedipine– Cimetidine	Melt-quench	Molar fractions 0.1–0.9, at an interval of 0.1	DSC thermograms of both freshly prepared samples and stored sample (increased *T*_g_, and lack of crystallization and melting endotherms)	The molar fractions of cimetidine were 0.3–0.9 (equal to Nifedipine–Cimetidine from 2.3:1 to 1:9).	N (only for the sample at the 1:1 molar ratio)	[77]
Nifedipine– Paracetamol	Melt-quench	Molar fractions 0.1–0.9, at an interval of 0.1	Phase diagrams to determine the eutectic point	1:1.5	Y	[73]
Nimesulide– Carvedilol	Melt-quench	Molar fractions 0.1–0.9, at an interval of 0.1	DSC thermograms of both freshly prepared samples and stored sample (increased *T*_g_, and lack of crystallization and melting endotherms)	The molar fractions of carvedilol were 0.3–0.8 (equal to Nimesulide–Carvedilol 2.3:1 to 1:4).	N (only for the sample at the 1:1 molar ratio)	[77]
Ofloxacin– Tryptophan	Freeze-drying	Molar ratios of 1:1, 1:2, 1:3, and also weight fractions 0.5–0.95	Kinetic solubility measurements of drug for freeze-dried samples; Comparison of the experimental *T*_g_s to the theoretical *T*_g_s for the largest deviation	Best solubility was found at the molar ratio of 1:1.76; highest positive deviation in *T*_g_ values was also found at a molar ratio of 1:1.76.	N (only for the CAMS at the 1:1.76 molar ratio)	[82]
Paracetamol– Antipyrine	Melt-quench	Molar fractions 0.1–0.9, at an interval of 0.1	Thermal analysis during physical stability	1:2	Y	[83]
Paracetamol–Celecoxib	Melt-quench	Molar fractions 0.1–0.9, at an interval of 0.1	Phase diagrams to determine the eutectic point	1:1	Y	[73]
ROY*– Pyrogallol	Melt-quench	Weight fractions 0–100 wt %, at an interval of 5 wt %	Thermal analysis by DSC (lack of any endothermic or exothermic events)	Pyrogallol content 25–35 wt % (equal to the molar ratio ROY*:Pyrogallol from 1:0.69 to 1:1.11)	N	[75]
Simvastatin– Nifedipine	Melt-quench	Molar fractions 0.1–0.9, at an interval of 0.1	Physical stability; phase diagram	CAMS at the molar ratio of 2:1 to 1:2 were all stable for at least one year (Eutectic composition: 5.375:1)	Y	[84]
Ursolic acid– Piperine	Solvent evaporation	3:1, 2:1, 1.5:1, 1:1 and 1:2	Determination of the highest *T*_g_; physical stability	2:1 showed the highest *T*_g_; 1.5:1 was the most stable CAMS	Y	[85]
Valsartan– Nifedipine	Melt-quench	Weight ratios of valsartan/nifedipine at 90:10, 80:20, 80:30 (molar 1:1), 60:40, 50:50, 40:60	Physical stability; in vitro dissolution test	CAMS at all molar ratios were stable; CAMS at the weight fractions of 80:30, 80:20, and 90:10 showed better drug release of both drugs (equal to the molar ratio 1:1, 1:0.67, and 1:0.3)	Y	[86]

#### 6.1.2. Detection of the Glass Transition Temperatures (*T*_g_)

An increase in the primary glass transition temperature (*T*_g_α) is often linked to an increase in physical stability [9]. The highest *T*_g_α in carvedilol–benzoic acid, carvedilol–malic acid, and carvedilol–citric acid CAMS were observed at the 1.5:1, 2:1, and 2:1 molar ratios of drug–organic acids, respectively. The following physical stability study showed that the most stable CAMS were consistent with the CAMS showing the highest *T*_g_α [39].

In addition, the comparison between the experimental *T*_g_α and the theoretical *T*_g_α calculated by the Gordon–Taylor equation can act as an indicator for molecular interactions between drug and co-former in CAMS. The molar ratio with the largest positive deviation between the theoretical and experimental *T*_g_α is potentially the molar ratio with the strongest interactions, which is hypothesized to result in a prolonged physical stability compared to other molar ratios. It was confirmed that carvedilol–aspartic acid and carvedilol–glutamic acid CAMS (at a molar ratio of 1:1.5), which showed the largest positive deviation between the theoretical and experimental *T*_g_α, indeed were the most physically stable CAMS. When applying a fitting model on the theoretical and experimental *T_g_*α to determine the exact optimal molar ratio value, the highest *T*_g_α was found to be at a molar ratio of 1:1.46 for carvedilol–aspartic acid CAMS and at a molar ratio of 1:1.43 for carvedilol–glutamic acid CAMS [72].

Positive deviations between the theoretical and experimental *T*_g_α are based on (strong) molecular interactions between drug and co-former; i.e., the above-mentioned method cannot be applied to CAMS with no or only weak interactions. In carvedilol–tryptophan CAMS, the theoretical and experimental *T*_g_α values were consistent with each other, and an increase in the temperature of *T*_g_α could not be linked to a prolonged physical stability [76]. To determine the optimal molar ratio (with respect to physical stability) in carvedilol–tryptophan CAMS, the secondary glass transition temperature (*T*_g_β) was measured by dynamic mechanical analysis (DMA). Depending on the molar ratio, the *T*_g_β of the excess component could be detected. Using a fitting model, the optimal molar ratio for carvedilol–tryptophan was found at 34–52% drug molar fraction. Even though the determined optimal molar ratios are given at a broad range, this study highlighted the potentially high importance of considering the *T*_g_β for the preparation of physically stable CAMS. It should be noted that at least *T*_g_α is of a kinetic nature, and the experimental value is therefore dependent on the preparation method, the parameters of the analysis method (e.g., heating rate), and the analysis method itself.

#### 6.1.3. Relationship between CAMS and the Eutectic Behavior of Crystalline Drug and Co-Former

The most physically stable molar ratio for naproxen–indomethacin CAMS was found at 1.5:1, rather than the expected equimolar ratio employing a molar ratio screening process. The most stable molar ratio represented the eutectic composition of a crystalline physical mixture of naproxen–indomethacin [74]. It was suggested that the interaction between the two drugs, naproxen and indomethacin, are strongest at the eutectic composition in both the melt and the resulting solidified CAMS.

Subsequently, the physical stability of CAMS at different molar ratios to their eutectic behavior was further related to each other by constructing the phase diagram of binary physical mixtures of naproxen–indomethacin, nifedipine–paracetamol, and paracetamol–celecoxib [73]. The study showed that the most stable CAMS were found at molar ratios, where the binary physical mixtures of the drugs had their respective eutectic points, i.e., naproxen–indomethacin 1.5:1 (as also shown by [74]), nifedipine–paracetamol 1:1.5, and paracetamol–celecoxib 1:1.

The eutectic behavior of binary mixtures can serve as a screening tool to determine the optimal molar ratio of stable CAMS, but it is limited to thermally stable components, i.e., the method is not suitable for most amino acids as co-formers even though a eutectic mixture of glimepiride and arginine has been reported [87].

#### 6.1.4. Application of Multivariate Analysis to Analytical Data

X-ray powder diffractograms (XRPD) and Fourier-transform infrared (FTIR) spectra of freshly prepared and stored CAMS were evaluated by principal component analysis (PCA) to analyze the crystallinity and amorphicity change during storage [74]. Based on the XRPD dataset, PC scores plots separated the samples according to their drug crystallinity and amorphicity, and the results showed that the highest total amorphous fraction was observed for naproxen–indomethacin at the molar ratio of 1.5:1, indicating the optimal molar ratio. In addition, PCA on the FTIR dataset was performed to further confirm the results from the XRPD dataset. The resulting PCA scores plot clustered the stored samples into three groups. The PC-1 loading plot showed that crystalline indomethacin signals contributed to the positive part and crystalline naproxen signals contributed to the negative part, while PC-2 separated crystalline samples (in the positive part of PC-2) from co-amorphous samples. The results showed that the naproxen–indomethacin 1.5:1 sample was placed near the zero line of PC-1 scores and at low PC-2 scores, which implies the least crystallinity and was hence consistent with the results obtained by XRPD.

Similarly, a PCA was performed on the FTIR spectra of freshly prepared carvedilol–aspartic acid and carvedilol–glutamic acid CAMS to investigate the spectral differences among CAMS at different molar ratios [72]. Similar-shaped distribution patterns of CAMS were obtained at different molar ratios in the PCA score plots, as those seen in the *T*_g_α deviations at different molar ratios (see Section 6.1.2.). The molar ratio of carvedilol–amino acids 1:1.5 was found to be the optimal ratio considering the physical stability, rather than the equimolar CAMS.

Application of multivariate analysis on different spectra of CAMS can visualize minor changes during storage and the underlying interaction differences among samples at different molar ratios. To date, multivariate analysis has been used to trace changes during storage, to explain the interaction mechanism, and to further confirm the found optimal molar ratio [72,74,88,89,90]. More research focusing on multivariate analysis on CAMS needs to be done to investigate the predictability of physical stability of CAMS.

Apart from the essentially predictive methods mentioned above to determine the optimal molar ratio, other, more direct methods, such as long-term physical stability tests were used [78,81,83,84,85]. It is worth noting that the molar ratio optimization methods mentioned above mainly focus on achieving highest physical stability. It remains unclear how the optimal molar ratio can affect the third CQA, dissolution performance, compared to non-optimal molar ratios, as this was only (partly) investigated in three studies [44,83,86]. Thus, investigations of the effect of the optimal molar ratio on the dissolution performance of CAMS needs further investigation. Nevertheless, new screening methods to determine the optimal molar ratio with respect to physical stability at an early stage of the preparation of CAMS should be further implemented in future research.

## 7. Preparation Methods of CAMS

The amorphous form of a drug can be obtained from the crystalline state of the drug via two fundamentally different pathways, i.e., the kinetic and the thermodynamic pathway. The same two pathways also apply for the preparation of CAMS. Using the kinetic pathway, the loss of order of the crystalline structure of a drug is obtained by inducing crystal defects by mechanical activation. A common preparation method to induce crystal defects is milling. Using the thermodynamic pathway, the loss of order of the crystalline structure of a drug is obtained by dissolving the drug in a solvent or melting the drug, followed by rapid solvent evaporation or rapid cooling (quench–cooling), respectively, that do not allow the crystal structure of the drug to be formed [91].

Our review of the CAMS studies showed that more than half (53.5%) of the CAMS were prepared using the kinetic pathway (Figure 7a) and 88.7% of those CAMS were obtained by ball milling while only a minor part of CAMS was prepared by cryo-milling (Figure 7c). CAMS prepared by the thermodynamic pathway were in 48.8% of the cases obtained by melting and rapid cooling (melt-quench) (Figure 7b). Spray drying was used in 26.1% of the CAMS prepared by the thermodynamic pathway and 20.4% were obtained by solvent removal, which is a method commonly used to obtain (co-)crystals but also exploited for the formulation of CAMS [92]. Other preparation methods include for example inkjet-printing (thermodynamic pathway) [93], hot melt extrusion (mainly thermodynamic pathway) [94,95,96], in situ co-amorphization [97,98,99,100], co-precipitation (thermodynamic pathway) [101], and co-grinding (kinetic pathway; defined as manually grinded) [102], of which some are difficult to assign to a certain pathway (thermodynamic or kinetic).

Interestingly, in some cases, depending on the used preparation method, a drug and a co-former could form a CAMS or not. Table 2 gives an overview of different outcomes reported depending on the used preparation method or using different preparative parameters, e.g., milling time, within the same preparation method for the top five drugs.

**Table 2 pharmaceutics-13-00389-t002:** Reported formation of CAMS (or not), depending on the preparation method for the top five drugs used for CAMS: indomethacin, mebendazole, carvedilol, carbamazepine, and simvastatin. For carbamazepine, only CAMS upon ball milling were reported and are not included in this table. Y: Yes, N: No. LAG: Liquid-assisted grinding.

Drug	Amino Acid	Co- Amorphous (Y/N)	Preparation Method	Notes	Reference
Carvedilol	Aspartic acid	N	Ball milling		[29,41,54,103]
Carvedilol	Aspartic acid	Y	Spray drying	Dependent on solvent composition	[72,103,104]
Carvedilol	Aspartic acid	N	LAG		[103]
Carvedilol	Glutamic acid	N	Ball milling		[29,41,54,103]
Carvedilol	Glutamic acid	Y	Spray drying	Dependent on solvent composition	[72,103]
Carvedilol	Glutamic acid	N	LAG		[103]
Indomethacin	Histidine	N	Ball milling		[29,41,54,103]
Indomethacin	Histidine	Y	Spray drying		[105,106]
Indomethacin	Lysine	Y	Ball milling	60 min	[29,41,54,107,108]
Indomethacin	Lysine	N	Ball milling	60 min at 4 °C	[105]
Indomethacin	Lysine	Y	Spray drying		[105,106,107]
Indomethacin	Lysine	N	LAG	Crystalline salt was formed	[108]
Mebendazole	Phenylalanine	Y	Ball milling	60 min	[29,41,54,107]
Mebendazole	Phenylalanine	N	Ball milling	Up to 180 min at 4 °C	[109,110]
Mebendazole	Proline	Y	Ball milling	Up to 180 min at 4 °C	[110]
Mebendazole	Proline	N	Ball milling	60 min	[29,41,54]
Simvastatin	Tryptophan	Y	Ball milling		[29,41,54]
Simvastatin	Tryptophan	N	Spray drying		[111]

It can be seen in Table 2 that the drug carvedilol could not form a CAMS upon ball milling [29,41,54,103] or using liquid-assisted grinding (LAG) [103] with the amino acid co-formers aspartic acid or glutamic acid but could form a CAMS upon spray drying [72,103,104]. The same behavior using different preparation pathways could be seen for indomethacin with the amino acid co-former histidine, which became co-amorphous upon spray drying [105,106] but not upon ball milling [29,41,54,103]. Interestingly, the exact opposite could be shown for the drug simvastatin and the amino acid co-former tryptophan, which became co-amorphous after ball milling [29,41,54] but not after spray drying [111]. Indomethacin became co-amorphous with the amino acid co-former lysine upon spray drying [105,106,107] and not after LAG, which resulted in the formation of the crystalline salt [108]. Upon ball milling, it was shown to be parameter dependent whether indomethacin became co-amorphous with the amino acid co-former lysine, as it did become co-amorphous after 60 min of ball milling (unknown temperature) [29,41,54,107,108] but not upon ball milling in a cold-room for 60 min at 4 °C [105].

An odd behavior was also shown for the drug mebendazole with the amino acid co-formers phenylalanine and proline. Mebendazole became co-amorphous with the amino acid co-former proline after longer ball milling times at a lower temperature [110] compared to a shorter ball milling time at an undefined but presumably higher temperature [29,41,54]. However, the same experimental conditions led to the opposite outcome for mebendazole and the amino acid co-former phenylalanine, which became co-amorphous after 60 min of ball milling [29,41,54,107] and was not co-amorphous after 180 min of ball milling at 4 °C [109,110]. It is unclear which experimental conditions could have led to obtaining these different results for the drug mebendazole. From a mechanistic perspective, an increase in the ball milling time and milling at a lower temperature should increase the likelihood of formation of a CAMS compared to shorter milling times at higher temperatures, as seen for mebendazole and the co-former amino acid proline (see above). It should be noted that ball milling can also result in the formation of co-crystals instead of CAMS. During ball milling, the kinetically induced destruction of the crystal lattice competes with the thermodynamically induced recrystallization [43,112]. It was reported that the existence of a co-crystal to a given drug–co-former CAMS (here: carbamazepine and organic acids) negatively impacted the formation of a CAMS or its physical stability, i.e., resulting in an initial CAMS, which quickly recrystallized after preparation [43]. On a more general note, ball milling is more likely to result in small amounts of residual crystallinity compared to spray drying or other thermodynamic preparation methods, which could potentially lead to shorter physical stability of ball-milled CAMS compared to spray-dried CAMS [112,113,114]. Further examples of drug–amino acid mixtures forming CAMS (or not) are discussed in Section 7.1.

### 7.1. “Rules of Thumb” for the Preparation Method Ball Milling for Drug–Amino Acid CAMS

In 2016, a systematic study was performed using six different drugs, two basic, two neutral drugs, two acidic drugs, and 20 different amino acids (see Figure 3, the most commonly investigated CAMS consist of a drug and an amino acid) [29]. Ball milling for 60 min was performed on all 120 combinations. The amino acids could be divided into generally good and bad co-formers depending on the drug acidity. In this study, acidic amino acids, namely aspartic acid and glutamic acid, were generally reported as poor co-formers, not leading to CAMS even for basic drugs. However, as described above, other preparation methods could result in the formation of a CAMS using acidic amino acids. Hence, other studies were screened for the use of acidic amino acids and the formation of a CAMS. Only the discussed examples in Table 2 were found; i.e., carvedilol formed a CAMS with the acidic amino acids glutamic acid and aspartic acid upon spray drying (also shown in Table 3) [72,103,104].

Furthermore, the polar amino acids, asparagine, cysteine, glutamine, serine, threonine, and tyrosine were categorized as generally poor co-formers. However, there are also some exceptions to this “rule of thumb”, as shown in Table 4.

In both cases, the drug glibenclamide formed a CAMS with the polar amino acids serine [28,115,116] and threonine [28,115] upon cryo-milling. The effect of cryo-milling compared to ball milling on the formation of CAMS has previously been described [20].

Basic amino acids, namely arginine, histidine, and lysine, were reported to be suitable co-formers for acidic drugs upon ball milling [29]. Hence, other studies were screened for exceptions reporting acidic drugs that did not form a CAMS with any of the basic amino acids and exceptions reporting basic drugs that formed a CAMS with any of the basic amino acids (Table 5).

The finding that no formation of CAMS was reported for the acidic drug indomethacin with the basic amino acid co-formers histidine and lysine was discussed above (see Table 2). The acidic drug ibrutinib was not able to form a CAMS with the basic amino acids arginine and histidine upon LAG [36]. However, it could well be that a CAMS can be formed upon ball milling or spray drying. The drug glibenclamide did not form a CAMS with the basic amino acid lysine upon cryo-milling [28]. It may be suggested that a preparation method following the thermodynamic pathway could be evaluated for the formation of a CAMS using glibenclamide and lysine. Glimepiride did not form a CAMS with the basic amino acid arginine upon ball milling, melt-quench, or solvent evaporation, but interestingly, a supercritical antisolvent technique using supercritical carbon dioxide led to the formation of a CAMS [87].

On the other hand, the basic drugs cimetidine and mebendazole were able to form a CAMS upon ball milling with the basic amino acid arginine [117]. In contrast to the study by Kasten et al. (2016), ball milling was performed for 180 min in a cold room at 5 °C [117] instead of for 60 min, which could explain the different result obtained. However, mebendazole and arginine formed a two-phase system, which does not meet the definition of a CAMS used here.

Furthermore, it was found that neutral amino acids can generally be classified as possible co-formers, except for the amino acids alanine and glycine, which were categorized as poor co-formers [29]. For the neutral amino acids alanine and glycine, no exceptions to this “rule of thumb” could be found in other studies; i.e., no CAMS with the neutral amino acids alanine and glycine is yet reported. The other neutral amino acids, isoleucine, leucine, methionine, phenylalanine, proline, tryptophan, and valine should be further considered for screening purposes.

Overall, for the use of amino acids as co-formers, more systematic studies of the preparation method are necessary to draw conclusions on which drug and amino acids can form a CAMS depending on the preparation method.

### 7.2. The Use of Peptides Instead of Single Amino Acids as Co-Formers in CAMS

Drug–amino acids CAMS are the most commonly reported CAMS (see above, Figure 3). Depending on the choice of amino acid, the three CQA are differently affected. It was shown that some amino acids cannot form CAMS with the chosen drug using specific preparation methods, e.g., ball milling (see Section 7.1).

The first use of a dipeptide was the use of Aspartame, consisting of the two amino acids, aspartic acid and phenylalanine [109]. Aspartic acid is a poor co-former using ball milling as a preparation method (see above), whilst phenylalanine does not show an enhanced dissolution performance. The dipeptide outperformed the single amino acids with respect to co-formability, physical stability, and dissolution performance (the three CQA). This case study showed that the properties of the CAMS can be tailored to the desired performance by combining co-formers with different molecular interactions with the drug. This is also suspected for some copolymer-based ASD, where only certain monomer units are responsible for different aspects of the overall formulation performance [118].

In a follow up study, several individual amino acids and dipeptides were investigated for the basic drug mebendazole [110]. Here, the dipeptides tryptophan–phenylalanine, phenylalanine–tryptophan, aspartic acid–tyrosine, histidine–glycine, and proline–tryptophan were investigated as co-formers for mebendazole and compared to the individual single amino acids and physical mixtures of the amino acids forming the dipeptides. The dipeptides showed generally a good co-formability, and their dissolution performance was enhanced, however, to varying degrees and resulting in different dissolution profiles. The physical stability was enhanced in all CAMS using dipeptides. The sequence of the amino acids in the dipeptide (tryptophan–phenylalanine vs. phenylalanine–tryptophan) could not be linked to any differences in the CQA, which was confirmed in a further study using the amino acids glutamic acid and arginine and their dipeptides [119]. As these are just three studies, further investigation of dipeptides or longer amino acid sequences are necessary to understand their impact on the three CQA and to what extend dipeptides can be exploited to tailor the three CQAs toward the pharmacokinetic and pharmacodynamic needs of the drug.

## 8. Physical Stability of Investigated CAMS

As mentioned in the introduction, CAMS are formed to improve the physical stability of the amorphous form of the drug (Figure 1). Hence, the second CQA of a CAMS is their (improved) physical stability. Whilst this review will focus on the physical stability of CAMS, the chemical stability can be of interest, especially when using ball milling, which has been shown to be able to induce degradation. The reader is referred to several studies, which have investigated the chemical stability (either after preparation or after storage) [18,20,21,33,34,47,87,98,102,120,121,122,123,124,125,126].

As seen in Figure 8a, a slight majority of the studies have included physical stability tests (56.1%). Most of these studies, 32.9% of the total number of studies, were conducted at dry conditions; 18.7% of the total number of studies cover physical stability studies at various humidity levels. In humid conditions, the CAMS can potentially adsorb and absorb moisture, which will reduce the *T*_g_ and increase the molecular mobility of the CAMS. Increased mobility can lead to phase separation and recrystallization phenomena and thereby reduce the stability of the CAMS. In addition, temperature is another important factor with regard to storage conditions for physical stability tests. As shown in Figure 8b, more than half of the physical stability of CAMS (52.7%) was evaluated at ambient temperature, and 39.2% of physical stability tests were conducted at elevated temperatures. Even though elevated temperatures were commonly used to provide a stress condition to accelerate physical stability tests; a quantitative link between stress condition stability and normal storage stability is missing.

As recrystallization and the loss of physical stability happen over time, it is important to also consider the storage periods of the CAMS at any given condition. Recrystallization is defined as recrystallization into one or more of the individual crystalline components (Note: sometimes in hydrate forms or in different polymorphic forms compared to the starting crystalline form) and also recrystallization into a co-crystal. It can be seen in Figure 8c that a majority of the studies used a storage period length of 6 months or less (62.9%). However, longer storage period lengths are usually necessary to assess the stability of a CAMS, e.g., up to 3 years. Only 13.5% of the studies included physical stability data that were obtained over a storage period length of longer than 18 months (Note: some CAMS recrystallized during that period). The longest so far reported stability studies for CAMS lasted two years [77,107,127]. As a comparison, for polymer-based ASD, physical stability data for up to 25 years have been reported [128].

Not only is it of interest to investigate the physical stability of CAMS under different storage conditions for a longer storage period length but also the differences in the physical stability of differently prepared CAMS. As the preparation method has an influence on the formed CAMS [112], it could also affect the physical stability of the CAMS, as shown for the pure amorphous drug simvastatin [129].

## 9. In Vitro and In Vivo Performance of CAMS

Dissolution performance is the third CQA of a CAMS. As CAMS are intended to be formed for the purpose of high amorphous stability, but ultimately to increase drug dissolution and solubility and hence, potentially bioavailability, it is important to investigate the in vitro and in vivo performance of CAMS. The dissolution performance can be investigated with respect to the dissolution rate, the apparent drug solubility, the overall ability to obtain supersaturation, as well as the maintenance of the supersaturation. However, it is well known that there is not always a clear link between the in vitro dissolution performance and the in vivo performance (pharmacokinetic profile) of a given drug delivery system [130,131]. That could mean that a superior in vitro dissolution performance of a CAMS compared to the pure amorphous form and crystalline state of the drug is not guaranteed to translate into a better bioavailability or even in vivo dissolution, e.g., due to different conditions in vitro compared to in vivo [132]. Thus, it is very important to conduct in vivo pharmacokinetic studies for CAMS that show promising in vitro results.

Figure 9 shows that only 44.1% of the reviewed studies have conducted in vitro dissolution experiments for CAMS. The majority of the studies did not investigate this third CQA of CAMS but might have only focused on the co-formability and physical stability. From the reviewed studies, 39.2% have only investigated the in vitro dissolution performance, and only a small number of all studies, 4.9%, have investigated the dissolution performance of CAMS in vitro and pharmacokinetic profiles in vivo, providing a link between the in vitro and in vivo dissolution performance of CAMS.

The reader is referred to other comprehensive review articles focusing on the supersaturation ability of CAMS and their overall (in vitro) dissolution performance [112,130].

In Table 6, an updated overview is given of studies covering the in vivo performance of CAMS. All studies were performed in rats, apart from one, which investigated the bioavailability of a CAMS formulated as an oral film in healthy men [33]. None of the top five investigated drugs, indomethacin, mebendazole, carvedilol, carbamazepine, or simvastatin were tested for their in vivo performance in a CAMS. Even though most of the data are available for these drugs, in vivo performance is missing so far. These five drugs were mainly screened and investigated for their other CQA and/or only in vitro dissolution performance. In contrast, the drug curcumin was in total investigated three times for its in vivo performance as a CAMS. The drug curcumin is categorized as a BCS Class IV drug and shows not only poor solubility and dissolution properties but also poor permeability. The idea of the formation of a CAMS consisting of a BCS Class IV drug, here curcumin, and a permeability enhancer as a co-former was shown to be promising [133]. In most cases, an enhanced bioavailability due to enhanced dissolution properties could be demonstrated in the studies including in vivo investigations. A non-significant increase in the area under the curve (AUC) of the plasma concentration–time curve and thereby a non-significant increase in the bioavailability of a CAMS compared to the crystalline state of the pure drug was found in a few studies [122,125,132]. Based on the limited in vivo data available for CAMS, the in vivo performance and the link between in vitro dissolution and in vivo performance requires further investigation as already previously discussed in a review article by Shi et al. 2019 [112].

Furthermore, only two studies have included molar optimization (as a part of co-formability), physical stability, and in vivo performance, hereby including all three CQA [85,86]. These studies included a CAMS consisting of ursolic acid and piperine [85] and the drug–drug CAMS valsartan and nifedipine [86]. Thus, there is not only a lack of studies investigating the in vivo performance of CAMS but also a lack of studies that rationally investigate all three CQA for a CAMS.

Apart from pharmacokinetic parameters, also the pharmacodynamic performance of CAMS was investigated in a few studies. The influence on the systolic blood pressure of the CAMS irbesartan and atenolol (1:1 molar ratio) was investigated in rats and resulted in a significant decrease compared to the crystalline physical mixtures [134]. For the glimepiride–arginine CAMS (1:1 molar ratio), the therapeutic efficacy was measured by the blood sugar level in rats. A better therapeutic efficacy was found compared to the pure drug glimepiride [87]. Using mice, the PANC-1 tumor growth was investigated for the CAMS curcumin–artemisinin (1:1) and was shown to inhibit the tumor growth by 62% [135] (see also Table 6 for pharmacokinetic in vivo study). An ex vivo study concluded on the flux of the CAMS ritonavir–lopinavir (molar ratio 2:1) using rats. A 10-fold increase in the flux of ritonavir and a 3-fold increase for lopinavir was found for the CAMS formulation compared to the pure crystalline drugs [136].

**Table 6 pharmaceutics-13-00389-t006:** Overview of studies investigating in vivo plasma–concentration profiles (pharmacokinetic profiles) of CAMS. AUC: area under the curve. C_max_: maximum concentration. T_max_: time point of maximum concentration. T_½_: halftime.

Drug	Co-Former	Molar Ratio	In Vitro Study	In Vivo Study	Reference
Atorvastatin Calcium	Nicotinamide	1:1	Intrinsic and powder dissolution rate enhanced compared to physical mixture	Rats: CAMS increased C_max_ by 2.25-fold and AUC by 1.72-fold compared to the crystalline drug;	[49]
Curcumin	Artemisinin	1:1	Enhanced intrinsic dissolution rate compared to crystalline curcumin	Rats: CAMS showed a C_max_ of 1005 µg/mL and an AUC of 24.7 µg*h/mL; crystalline CUR could not be detected in the plasma;	[137]
Ritonavir	Quercetin	1:2	-	Rats: CAMS improved C_max_ by 1.26-fold and T_max_ decreased by 2 h compared to the crystalline drug; however, no significant enhancement in oral bioavailability was found (AUC);	[132]
Talinolol	Naringin	1:1	-	Rats: CAMS improved C_max_ by 8.6-fold, T_max_ decreased by 1.5 h and AUC improved by 5.4-fold compared to the crystalline drug;	[124]
Olanzapine	Ascorbic Acid Citric Acid Tartaric Acid	1:1, 1:2	Enhanced dissolution rate	Healthy men: CAMS has 115.83% bioavailability compared to a marketed product and showed a faster disintegration; Note: the CAMS was formulated as an oral film with polymeric excipients.	[33]
Atenolol	Hydrochlorothiazide	1:1	Enhanced intrinsic dissolution rate	Rats: CAMS improved C_max_ by 7.3-fold compared to crystalline drug (hydrochlorothiazide), by 2.8-fold compared to the amorphous drug and by 1.7-fold compared to the physical mixture. AUC was increased by 3.4-, 2.6- and 1.4-fold compared to the crystalline drug, the amorphous drug, and the physical mixture.	[138]
Loratadine	Citric Acid	1:1	Enhanced solubility and dissolution rate compared to the crystalline drug	Rats: CAMS improved C_max_ by 2.59-fold compared to the crystalline drug. T_½_ decreased by 2.5 h. AUC improved by 2.45-fold. Enhanced absorption of loratadine.	[139]
Naproxen	Arginine	1:1	Enhanced intrinsic dissolution rate compared to the crystalline drug	Rats: CAMS improved C_max_ by 2.15-fold and AUC by 1.5-fold compared to the crystalline drug. For the crystalline salt of the CAMS, no increase in bioavailability was seen, even though the in vitro performance was enhanced.	[140]
Curcumin	Artemisinin	1:1	-	Rats: CAMS showed a C_max_ of 1.23 µg/mL, a T_max_ of 30 min, and an AUC of 3.68 µg·h/mL. The crystalline drug curcumin could not be detected.	[135]
Curcumin	Piperine	1:1	Enhanced powder dissolution rate and higher supersaturation compared to crystalline drug	Rats: CAMS improved C_max_ by 2.64-fold for curcumin and by 2.41-fold for piperine. The AUC was improved by 2.16-fold and 1.92-fold for the individual drugs.	[133]
Docetaxel	Myricetin (natural p-Gp inhibitor)	1:1	Enhanced intrinsic and powder dissolution rate compared to crystalline drug	Rats: CAMS improved C_max_ by 2.3-fold and AUC by 1.7-fold compared to the physical mixture. C_max_ and AUC increased by 1.5- and 2.3-fold respectively compared to the amorphous drug docetaxel. CAMS improved C_max_ and AUC by 3.9-fold and 3.13-fold, respectively, compared to the crystalline drug docetaxel. Thus, the bioavailability of docetaxel compared to the crystalline drug is 313%. CAMS improved C_max_ and AUC for the crystalline drug myricetin by 2.1-fold and 1.9-fold, respectively.	[141]
Docetaxel	Bicalutamide	1:1	Enhanced dissolution rate for both drugs, but supersaturation only achieved for docetaxel	Rats: CAMS improved C_max_ by 8.8-fold and AUC by 11.8-fold for docetaxel compared to the crystalline drug. C_max_ improved by 3.3-fold and AUC by 3.2-fold for bicalutamide compared to the crystalline drug.	[142]
Valsartan	Nifedipine	Weight ratio: 80:30 (2.12:1 molar ratio)	Enhanced dissolution rate and supersaturation for both drugs	Rats: CAMS improved C_max_ by 3.63-fold and AUC by 1.44-fold compared to the crystalline drug nifedipine. CAMS improved C_max_ by 2.2-fold and AUC by 1.4-fold compared to the crystalline drug valsartan.	[86]
Ibrutinib	Saccharin	1:1	Enhanced dissolution rate and supersaturation	Rats: CAMS increased C_max_ by 2.9-fold compared to the crystalline drug. AUC, T_max_, and T_½_ were not significantly different.	[122]
Ibrutinib	Oxalic Acid (and microcrystalline cellulose)	1:1:1	Enhanced dissolution rate	Rats: CAMS improved C_max_ by 1.49-fold and AUC by 1.48-fold compared to the crystalline drug.	[36]
Ursolic acid	Piperine	1.5:1	Enhanced dissolution rate and supersaturation	Rats: CAMS improved C_max_ by 4.9-fold and AUC by 5.77-fold compared to crystalline ursolic acid.	[85]
Sacubitril	Valsartan (additionally with lactose monohydrate or microcrystalline cellulose)	1:1; weight ratio co-amorphous to excipient 1:3	Enhanced dissolution rate and supersaturation of the co-amorphous formulations	Rats: CAMS were compared to the marketed formulation Entresto^®^. A 1.54-fold higher AUC was found for valsartan and a 3.56-fold higher AUC was found for the sacubitril derivate in the CAMS with lactose monohydrate. A 1.39-fold higher AUC was found for valsartan and a 1.25-fold higher AUC was found for the sacubitril derivate in the CAMS with microcrystalline cellulose. However, there was decreased bioavailability for the CAMS with microcrystalline cellulose compared to the binary CAMS.	[121]
Atorvastatin	Naringin	1:1	Enhanced dissolution rate and supersaturation (but fast precipitation after 30 min)	Rats: Melt-quench CAMS improved C_max_ by 1.73-fold compared to the physical mixture. AUC was not found to be significantly different. Solvent evaporated CAMS improved C_max_ by 1.73-fold and AUC by 3.3-fold compared to the physical mixture.	[125]

Other in vitro studies inter alia including permeability studies, cytotoxicity studies, and aerosolization experiments are summarized in Table 7. These in vitro studies were performed to investigate other relevant effects that the CAMS might have depending on the composition and the drug delivery route (oral, dermal, or pulmonary). Especially, as mentioned before, permeability studies are of interest for CAMS containing a drug from BCS Class IV. However, in the above-mentioned studies, molar ratio optimization only was performed in two studies [44,85].

Overall, the limited data available on the in vivo performance of CAMS calls for more research in this area. Furthermore, for BCS Class IV drugs, the formation of a CAMS with a permeation enhancer as a co-former might be an interesting approach to overcome this further limitation of BCS Class IV drugs. Last but not least, also other drug delivery routes than the oral drug delivery route should be investigated more, such as dermal drug delivery and pulmonary drug delivery of CAMS.

## 10. Preparation of Ternary CAMS

The addition of a third component into a CAMS to form ternary amorphous drug delivery systems has recently gained interest, and apart from binary CAMS, some ternary systems are also reported in the literature (Figure 10). The third component, which could be a polymer, surfactant, or other small molecule, is added into the binary CAMS in order to optimize the CQA of CAMS (Table 8). The third component can be added to the binary CAMS after the formation of the CAMS (external third component), for example as a precipitation inhibitor, or it can be incorporated into the CAMS simultaneously with the other components (internal third component).

First, 10 wt % hydroxyl propyl methyl cellulose (HPMC) was added into carvedilol–aspartic acid CAMS to optimize the dissolution behavior of CAMS and it was found that the addition of HPMC improved the dissolution performance compared to the corresponding CAMS by reducing the very high initial dissolution rate and maintaining super-saturation for a longer time (CAMS tend to precipitate fast, see Figure 1, referring to the reviews mentioned in Section 9) [104]. Thus, the preparation of ternary drug–co-former–polymer CAMS showed potential by combining the dissolution advantages of both CAMS and polymer-based ASD, showing a lighter “spring” and an enhanced “parachute” effect compared to the corresponding binary CAMS (see Figure 1). The dissolution behavior was attributed to the fact that small molecules (here the co-former) can drastically accelerate the initial dissolution rate of the drug due to molecular interactions, whilst larger molecules (here polymer) can act as precipitation inhibitor and/or release rate-modulator to maintain supersaturation.

Different co-formulated surfactants were investigated with naproxen–arginine and naproxen–lysine CAMS prepared by freeze-drying [57]. The results indicated that the formulation and the CQA of the CAMS were affected by the type of surfactant. In addition, sodium lauryl sulfate was added into glibenclamide–amino acids CAMS to improve drug permeability [143]. It was found that both dissolution and permeation of the drug increased for the glibenclamide–arginine–sodium lauryl sulfate ternary system compared to the corresponding binary CAMS, but no significant improvement was observed in glibenclamide–serine CAMS with the addition of sodium lauryl sulfate. This indicates that more investigations on different CAMS and different permeation enhancers are required to obtain a comprehensive guideline for designing ternary systems to achieve permeability improvement. In general, a rational choice of the additional third component based on the binary CAMS is warranted.

**Table 7 pharmaceutics-13-00389-t007:** Overview of additional in vitro performance studies conducted on CAMS. NGI: Next-Generation Impactor. PVP: polyvinylpyrrolidone (polymer). PAMPA: parallel artificial membrane permeability assay.

Drug	Co-Former	Molar Ratio	In Vitro Study	Outcome	Reference
Sacubitril	Valsartan (also with lactose monohydrate and microcrystalline cellulose)	1:1 (with excipient weight ratio: 1:1, 1:2, 1:3, 1:4)	Permeation test	CAMS impaired the permeation of the drugs, i.e., the flux decreased by 27.8% for sacubitril and 31.0% for valsartan.	[121]
Hydrochlorothiazide (HCT)	Arginine (L and D form) (and plus PVP)	1:1 (with PVP in 1:1 weight ratio)	Permeation studies with PAMPA membranes; Permeation studies with MCDKII cellular barriers	Highest cumulative amount of permeated HCT for HCT:L-Arginine (over 2-fold compared to crystalline drug) using PAMPA membranes. Improved permeation of BCS class IV drug HCT even though no specific permeation-enhancing effect for the excipients could be found using MCDKII cellular barriers.	[144]
Budesonide	Theophylline	1:1	Aerosolization performance by NGI	Higher fine particle fraction compared to nanosuspensions and co-deposition of budesonide and theophylline in the aerodynamic assessment.	[145]
Simvastatin	Leucine Tryptophan Lysine	1:1	Aerosolization performance by NGI	Simvastatin-Leucine showed the best aerosol performance, followed by CAMS with tryptophan and lysine.	[111]
Chloramphenicol	Arginine Cysteine Glycine Leucine	1:1	Antimicrobial activity; Oxygen species detection	Chloramphenicol maintained its microbiological activity in CAMS with amino acids, i.e., the amino acids did not interfere with the microbiological activity of chloramphenicol	[146]
Ciprofoloxacin	Tartaric acid	2:1, 1:1, 1:2, 1:3	Antimicrobial activity	With tartaric acid as a co-former, the CAMS was more potent compared to the drug alone toward *P.aeruginosa* biofilms, which was described as a synergistic effect of the CAMS.	[147]
Ibrutinib	Oxalic acid (also with microcrystalline cellulose)	1:1 (:1)	Cytotoxicity assay	Ibrutinib–oxalic acid and microcrystalline cellulose as a CAMS reduced the side effects of the drug on the kidney (nephrotoxicity) and showed and improved antitumor effect.	[36]
Glibenclamide	Arginine Serine Arginine–sodium lauryl sulfate Serine-sodium lauryl sulfate	1:1; arginine-sodium lauryl sulfate (1:1:0.083 and 1:1:0.157) serine sodium lauryl sulfate (1:1:0.875 and 1:1:0.154)	Permeation study (PAMPA membranes)	Permeation was increased (AUC) by 7.2-, 5.7-, and 7.0- fold for CAMS with arginine, with arginine–sodium lauryl sulfate (low amounts) and arginine–sodium lauryl sulfate (high amounts), respectively, compared to the amorphous drug.	[143]
Azithromycin Tobramycin Ciprofloxacin	N-acetylcysteine	1:2 1:1.5 1:1	Aerosolization performance by NGI; Pseudomonas aeruginosa biofilm assay;	All CAMS showed a high fine particle fraction. The CAMS improved or at least maintained the antibiotic susceptibility and the inhibitory properties of N-acetylcysteine against P. aeruginosa biofilms.	[148]
Budesonide	Arginine	1:1	In vitro lung deposition test; Aerosolization performance by NGI	Aerosolization performance as well as lung deposition of budesonide improved with the co-former arginine.	[149]
Ursolic acid	Piperine	2:1, 1.5:1	In vitro permeability study across Caco-2 Cell Monolayers	Free piperine significantly increased the permeability of the drug. However, piperine in the CAMS exhibited a much lower level in permeability enhancement compared to its free form arising from the synchronized dissolution characteristic of the preparation.	[85]
Piroxicam	Citric Acid	1:1	In vitro skin permeation study	The CAMS demonstrated higher skin permeation than piroxicam alone or the physical mixture of the CAMS.	[150]
Kanamycin sulfate	Valine Methionine Phenylalanine Tryptophan	1:1	Aerosolization performance by NGI	All the CAMS improved the aerosolization performanc compared to the pure drug in the order methionine > tryptophan > phenylalanine > valine.	[88]
Glibenclamide	Arginine Serine Quercetine Arginine–sodium lauryl sulfate	1:1	Permeability studies conducted with MDCKII-MDR1 cells	The CAMS with arginine-sodium lauryl sulfate exhibited a 9-fold increase in permeating through the MDCKII-MDR1 cell layer as compared to the corresponding physical mixture. Permeability of the CAMS in the order of the co-former: serine < quercetine < arginine < arginine-sodium lauryl sulfate.	[116]
Talinolol	Naringin	1:1	In vitro single pass perfusion studies conducted on the ileum of Wistar rats	The permeability of talinolol was significantly increased in the presence of naringin due to the p-Gp inhibition effect by naringin.	[124]
Atenolol	Urea and PEG400	various	Skin permeation study	The supersaturated CAMS formulation showed higher permeability for mice skin than that of a supersatured drug formulation, due to the degree of supersaturation.	[44]
Curcumin	Piperine	1:1	Permeability study with caco-2-cell	The absorptive transport of curcumin was significantly enhanced by 2.67-fold compared to the pure crystalline drug, suggesting the CAMS can promote the intestinal absorption of CUR.	[133]
Acyclovir	Citric acid	1:10	Skin permeation study	The steady-state permeation flux of the drug in the CAMS was 2.06 µg/cm^2^/h, much higher compared to the crystalline pure drug (0.02 µg/cm^2^/h).	[151]

A third excipient was also incorporated into a CAMS formulation to optimize the preparation process and to achieve potential performance advantages by tableting and coating [67,94,97,100]. The addition of small amounts (5 wt %) of polyethylene oxide was demonstrated to hinder amorphous–amorphous phase separation effectively in the extrudates of indomethacin–cimetidine CAMS [94]. Moreover, the addition of PVP into indomethacin–arginine CAMS tablets could stabilize the supersaturation of indomethacin for a longer time compared to the tablet without PVP due to the precipitation inhibiting effect of PVP. The addition of PVP into ibuprofen–arginine CAMS tablets increased the initial drug release rate compared to tablets without PVP due to the strong interactions between drug and PVP [67]. Similarly, after coating with the Kollicoat^®^ protect polymer, the areas under the curves of the drug release of the coated indomethacin–arginine CAMS tablets increased by 30% compared to the respective uncoated formulations [97].

Despite these promising examples using ternary CAMS, potential risks should also be taken into consideration. A similar concept (i.e., adding a small molecule into ASDs to improve the performance of ASDs) has been applied in the polymer ASD research [96,152]. However, it was recently discovered that the addition of saccharin or tryptophan resulted in a decreased physical stability of carbamazepine–PVP ASD in both predictions and experiment validations [56]. This work provided another view on the formulation of ternary systems. As observed in ASDs, it is worthwhile to consider that the preparation of ternary systems based on CAMS might not only offer opportunities but could also bear risks. Thus, there are still research questions that remain unanswered for ternary amorphous drug delivery formulations. The influence of the addition of a third component on the three CQAs compared to the corresponding binary CAMS is necessary to be evaluated before making a conclusion if and how a desired ternary amorphous drug delivery formulation can be obtained.

**Table 8 pharmaceutics-13-00389-t008:** Overview of reported ternary amorphous systems. HME: hot melt extrusion. HPMC: hydroxyl propyl methyl cellulose (polymer). PVP: polyvinylpyrrolidone. PEO: polyethylene oxide (polymer). TPGS: Tocophersolan.

Binary CAMS	The Additional Third Component	Type of the Additional Component	Preparation Method(s)	Reasons for the Addition	Outcomes (Compared to the Corresponding Binary CAMS)	Reference
Carbamazepine–Citric acid	Arginine	Small molecule	Ball milling	To design a stable co-amorphous system with an elevated *T*_g_	Significant increase in the *T*_g_ value; Improvement of dissolution behavior and physical stability	[153]
Carvedilol–Aspartic acid	Eudragit^®^ L 55	Polymer	Coating and in situ amorphization	As a coating dispersion	-	[100]
Carvedilol–Aspartic acid	HPMC	Polymer	Spray drying	To optimize the dissolution behavior	Improvement of dissolution behavior by reducing the initial dissolution rate and maintaining supersaturation for a longer time	[104]
Ciprofloxacin–Tartaric acid	Silica-coated silver nanobeads and NaHCO_3_	Other	Spray drying	As an external layer for co-amorphous powder coating	Disruptive effect on rheological properties	[147]
Ezetimibe–Lovastatin	Soluplus^®^	Polymer	Spray drying	To improve the poor dissolution characteristic	Improvement of the dissolution behaviors of both drugs	[78]
Ezetimibe–Lovastatin	PVP K30	Polymer	Spray drying	To improve the poor dissolution characteristic	Significant improvement of the dissolution behavior of only one drug	[78]
Ezetimibe–Lovastatin	PVP VA64	Polymer	Spray drying	To improve the poor dissolution characteristic	Significant improvement of the dissolution behavior of only one drug	[78]
Ezetimibe–Lovastatin	HPMC	Polymer	Spray drying	To improve the poor dissolution characteristic	Significant improvement of the dissolution behavior of only one drug	[78]
Ezetimibe–Simvastatin	Kollidon^®^ VA64	Polymer	Melt-quench	To verify feasible applications of the developed methods	-	[154]
Ezetimibe–Simvastatin	Kollidon^®^ VA64	Polymer	Melt-quench	To hinder the re-crystallization	Improvement of the physical stability at elevated temperature conditions (T = 373 K)	[79]
Flutamide–Bicalutamide	Poly(methyl methacrylate-*co*-ethyl acrylate)	Polymer	Melt-quench	To stabilize two drugs mixed	Inhomogeneity of the sample	[155]
Flutamide–Bicalutamide	PVP	Polymer	Melt-quench	To stabilize two drugs mixed	Sample homogeneity; Inhibition of the re-crystallization	[155]
Glibenclamide–Arginine	Sodium lauryl sulfate	Surfactant	Cryo-milling	To act as an absorption enhancer for permeability improvement	Improvement of dissolution and permeability of the drug	[116,143]
Glibenclamide–Serine	Sodium lauryl sulfate	Surfactant	Cryo-milling	To act as an absorption enhancer for permeability improvement	No significant improvement on the dissolution and permeability	[143]
Hydrochlorothiazide– Arginine	PVP	Polymer	Cryo-milling	Not mentioned	Improvement of drug dissolution behavior; Decrease on the drug permeation behavior	[143]
Ibrutinib– Oxalic acid	Microcrystalline cellulose	Polymer	Ball milling	As an effective crystal growth inhibitor	Improvement of solubility and dissolution rate; Improvement of physical stability	[36]
Ibuprofen–Arginine	Mannitol+ PVP K30	Small molecule and polymer	Tableting compaction	As tablet compositions	The addition of PVP increased the initial drug release rate	[67]
Ibuprofen–Arginine	Xylitol+ PVP K30	Small molecule and polymer	Tableting compaction	As tablet compositions	-	[67]
Indomethacin–Arginine	Co-povidone	Polymer	HME	To investigate the need for an addition of a polymer in the co-amorphous system preparation by HME	The co-amorphous formulations could be achieved with or without polymer; Enhanced dissolution behavior	[95]
Indomethacin–Arginine	Mannitol+ PVP K30	Small molecule and polymer	Tableting compaction	As tablet compositions	The addition of PVP showed precipitation inhibitory effect	[67]
Indomethacin–Arginine	Xylitol+ PVP K30	Small molecule and polymer	Tableting compaction	As tablet compositions	-	[67]
Indomethacin–Arginine	Kollicoat^®^ Protect	Polymer	Coating	To investigate whether polymer coating of co-amorphous formulations is possible without inducing recrystallization	Coating of a co-amorphous formulation is possible without inducing recrystallization; Improvement of the drug release behavior	[97]
Indomethacin–Cimetidine	PEO	Polymer	HME	To investigate the effects of the addition of low amounts of polymer on the processability during HME	Inhibition behavior of amorphous–amorphous phase separation; Decrease in melt viscosity	[94]
Indomethacin–Citric acid	PVP	Polymer	Solvent evaporation	To prevent self-association between these two small molecules and thus to enhance their mutual miscibility	Enhancement of the mutual miscibility between two small molecules, but the ability is sensitive to PVP concentration	[25]
Naproxen–Arginine	Sodium dodecyl sulfate	Surfactant	Freeze-drying	To increase the solubility of drug in the starting solution for freeze-drying; To investigate the influence of the surfactant types	Formation of a heterogeneous system; Improvement of sample physical stability at certain concentration	[57]
Naproxen–Arginine	Pluronic F-127	Surfactant	Freeze-drying	To increase the solubility of drug in the starting solution for freeze-drying; To investigate the influence of the surfactant types	Formation of a homogeneous system	[57]
Naproxen–Arginine	Polyoxyethylene (40) stearate	Surfactant	Freeze-drying	To increase the solubility of drug in the starting solution for freeze-drying; To investigate the influence of the surfactant types	Formation of a homogeneous system; Improvement of sample physical stability	[57]
Naproxen–Arginine	Tween 20	Surfactant	Freeze-drying	To increase the solubility of drug in the starting solution for freeze-drying; To investigate the influence of the surfactant types	Formation of a heterogeneous system; Improvement of sample physical stability at certain concentration	[57]
Naproxen–Arginine	TPGS 1000	Surfactant	Freeze-drying	To increase the solubility of drug in the starting solution for freeze-drying; To investigate the influence of the surfactant types	Formation of a heterogeneous system; Improvement of sample physical stability at certain concentration	[57]
Naproxen–Arginine	Proline	Small molecule	Ball milling	To achieve an additional improvement of the dissolution rate	Improvement of dissolution rate for Naproxen–Arginine amorphous salt	[156]
Naproxen– Indomethacin	Naproxen–sodium	Small molecule	Melt-quench	To optimize the physicochemical properties	Improvement of physical stability	[81]
Naproxen– Lysine	Sodium dodecyl sulfate	Surfactant	Freeze-drying	To increase the solubility of drug in the starting solution for freeze-drying; To investigate the influence of the surfactant types	Formation of a homogeneous system	[57]
Naproxen– Lysine	Pluronic F-127	Surfactant	Freeze-drying	To increase the solubility of drug in the starting solution for freeze-drying; To investigate the influence of the surfactant types	Formation of a homogeneous system; Improvement of sample physical stability	[57]
Naproxen– Lysine	Polyoxyethylene (40) stearate	Surfactant	Freeze-drying	To increase the solubility of drug in the starting solution for freeze-drying; To investigate the influence of the surfactant types	Formation of a homogeneous system; Improvement of sample physical stability	[57]
Naproxen– Lysine	Tween 20	Surfactant	Freeze-drying	To increase the solubility of drug in the starting solution for freeze-drying; To investigate the influence of the surfactant types	Formation of a homogeneous system; Improvement of sample physical stability	[57]
Naproxen– Lysine	TPGS 1000	Surfactant	Freeze-drying	To increase the solubility of drug in the starting solution for freeze-drying; To investigate the influence of the surfactant types	Formation of a heterogeneous system; Improvement of sample physical stability	[57]
Naproxen–Meglumine	Kollidon VA64	Polymer	Reactive melt extrusion	To combine the advantages of both salts and amorphous solid dispersions for enhancing the solubility and dissolution rates	The presence of a polymer did not interfere with the salt formation; Improvement of the dissolution properties and the physical stability compared to drug–polymer systems	[80]
Naproxen– Meglumine	Kollidon K30	Polymer	Reactive melt extrusion	To combine the advantages of both salts and amorphous solid dispersions for enhancing the solubility and dissolution rates	The presence of a polymer did not interfere with the salt formation; Improvement of the dissolution properties and the physical stability compared to drug–polymer systems	[80]
Naproxen–Meglumine	Soluplus	Polymer	Reactive melt extrusion	To combine the advantages of both salts and amorphous solid dispersions for enhancing the solubility and dissolution rates	The presence of a polymer did not interfere with the salt formation; Improvement of the dissolution properties and the physical stability compared to drug–polymer systems	[80]
Naproxen–Tryptophan	Proline	Small molecule	Ball milling	To achieve an additional improvement of the dissolution rate	Successful formation of amorphous mixture (while a small remaining degree of crystallinity observed in Naproxen–Tryptophan binary system); Improvement of dissolution rate	[156]
Nateglinide–Metformin hydrochloride	Magnesium aluminometasilicate	Small molecule	Spray drying	To improve flowability of spray-dried powder	Improvement of flow properties and enhanced compressibility; Improvement of solubility and dissolution; The formation of spherical microstructured particles; Enhancement of physical stability	[157]
Olmesartan medoxomil–Hydrochlorothiazide	HPMC	Polymer	Solvent evaporation	To inhibit the deleterious interactions	Inhibition of co-crystallization but no significant improvement on the dissolution rate	[158]
Sacubitril–Valsartan	Lactose monohydrate	Small molecule	Spray drying	As an inert carrier	A slight decrease of solubility of both drugs; Delay of phase transformation; Improvement of in vivo bioavailability; Improvement of powder properties for compressibility	[121]
Sacubitril–Valsartan	Microcrystalline cellulose	Polymer	Spray drying	As an inert carrier	A slight decrease of solubility of both drugs; Delay of phase transformation; Decrease on in vivo bioavailability; Improvement of powder properties for compressibility	[121]

## 11. Conclusions and Future Perspectives

In general, CAMS have been widely established as a promising drug delivery system to improve the dissolution performance and physical stability of poorly water-soluble drugs in an amorphous form, offering the benefit of high drug loading due to the lower amounts of required excipients (co-formers). Although the research on CAMS has continuously increased in the past decades and nearly 200 studies have been published, there are not only still unexplored research areas but also too few systematic investigations of CAMS. Further research will only lead to an optimization of the three CQA of CAMS but also bring CAMS closer to the market.

Co-former selection, choice of the drug to co-former molar ratio, and preparation methods are three crucial aspects regarding the first CQA, the co-formability of CAMS. This review revealed that in most of the investigated CAMS, co-former screening has not been considered, and the used molar ratios have been chosen without optimization. In a few cases, modern in silico tools have been applied for co-former selection, highlighting a potential to achieve a more efficient research approach in this field. Thus, more effort needs to be put on systematic studies and predictive screening approaches in this field, also taking advantage of the large datasets obtained from case studies. Furthermore, nearly half of the research studies in which the drug to co-former molar ratio was optimized indicated that the optimal molar ratio was not the 1:1 molar ratio, as often expected and used. This emphasizes the importance of molar ratio optimization in order to improve the three CQA of CAMS.

As shown in this review, more data also need to be collected considering the second CQA, the physical stability of CAMS. Long-term stability studies at various conditions (accelerated at high temperatures and high humidity) should be conducted. The absence of phase separation and recrystallization phenomena of CAMS at the optimal molar ratio of drug and co-former after long storage periods can potentially prove to be an advantage over the use of other solid dispersion techniques, such as polymer-based ASDs.

Dissolution performance (third CQA) of CAMS has only in very few cases investigated the correlation between in vivo and in vitro performance, indicating that the link between in vitro and in vivo performances still needs to be more firmly tackled to draw convincing conclusions. Furthermore, the existing drug state and possible phase changes (such as liquid–liquid phase separation) in the physiological environment remains unexplored; however, this may show a large influence on the dissolution performance of CAMS. In addition, the in vivo performance of CAMS should also be compared to other ASDs such as polymer- and mesoporous silica-based ASDs.

More research is also necessary in the new field of ternary CAMS aiming to optimize the three CQAs, especially the potentially missing precipitation inhibition in binary CAMS just consisting of the drug and the co-former (third CQA). Optimization of the three CQA has also been shown using peptides instead of amino acids to tailor the performance of CAMS. As these results are based on a few case studies showing a superior performance of dipeptides compared to the single individual amino acids, more systematic research also in this area is necessary.

Not only are CAMS a promising drug delivery strategy for BCS Class II drugs, but also for BCS Class IV drugs. Recent studies have shown that choosing a permeability enhancer as the co-former to a BCS Class IV drug can increase the overall bioavailability of these drugs. More studies on the permeability of CAMS and the use of BCS Class IV drugs are necessary, as the permeability enhancement could potentially be a major advantage over ASDs.

The use of CAMS has also shown promise, that could be realized in future studies, in other administration routes of drugs, such as pulmonary and buccal administration, instead of oral administration.

Overall, CAMS are a promising drug delivery strategy for BCS Class II and IV drugs; however, more research is necessary to fully exploit and optimize CAMS to achieve their full potential.

## Figures and Tables

**Figure 1 pharmaceutics-13-00389-f001:**
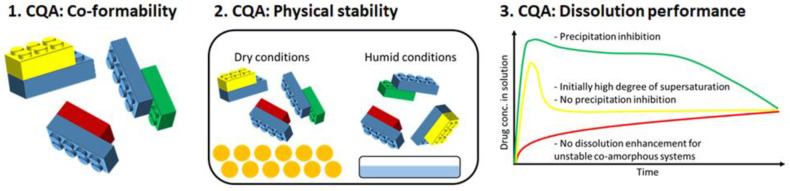
The three critical quality attributes (CQA) of co-amorphous drug delivery systems (CAMS): (**1**). co-formability, (**2**). physical stability, and (**3**). dissolution performance. The blue bricks represent a drug molecule and the green, red, and yellow bricks indicate co-formers. The yellow circles indicate e.g., silica gel.

**Figure 2 pharmaceutics-13-00389-f002:**
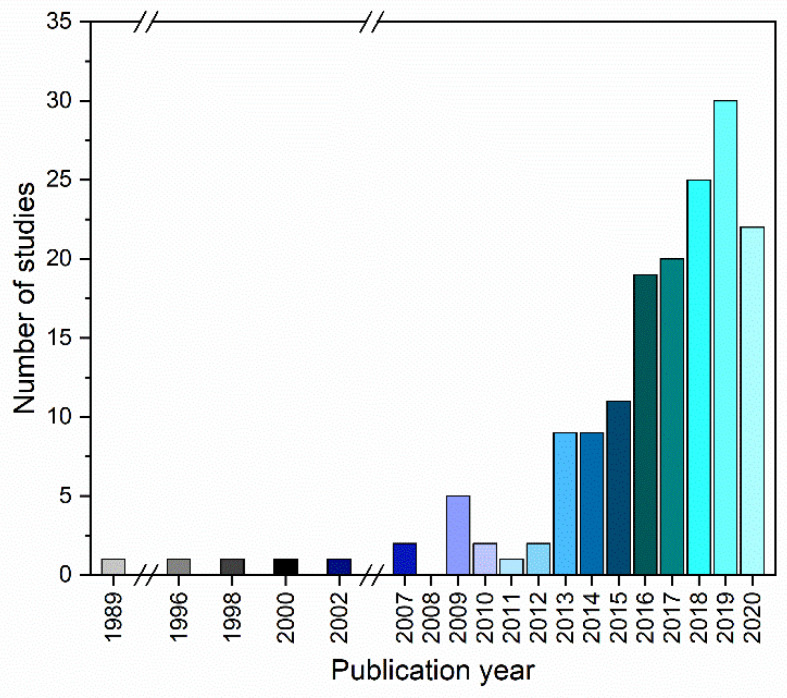
Column diagram showing the number of studies by year from 1989 to the year 2020.

**Figure 3 pharmaceutics-13-00389-f003:**
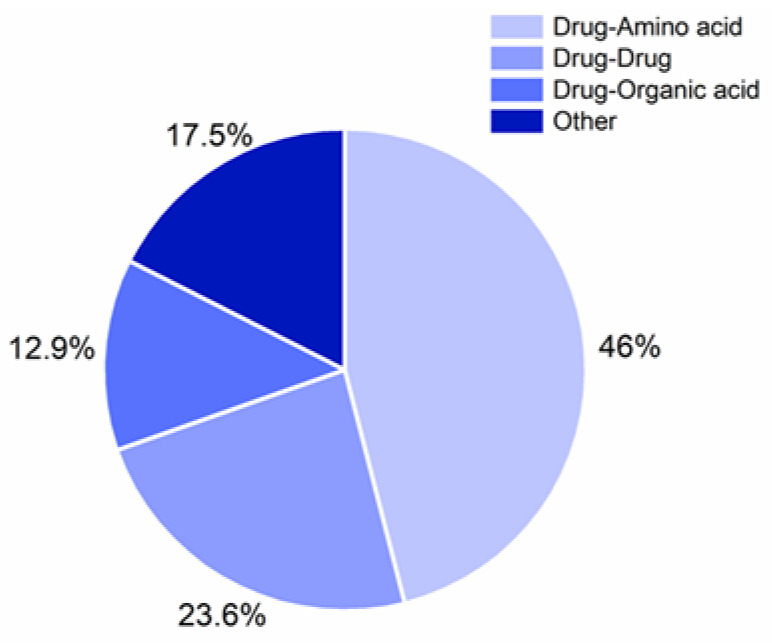
Percentage of the different classes of co-amorphous drug delivery systems (CAMS) reported in the literatures.

**Figure 4 pharmaceutics-13-00389-f004:**
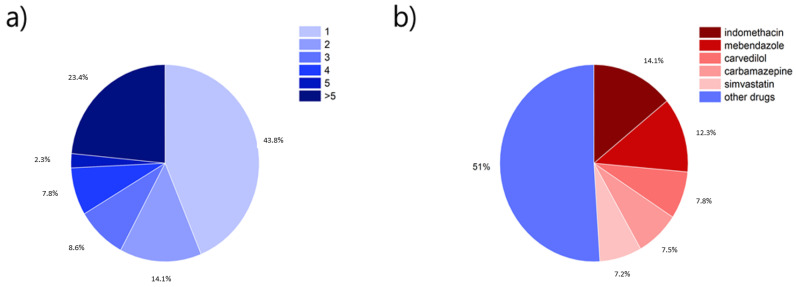
(**a**) Percentage of how often a specific drug has been reported in different studies (1, 2, 3, 4, 5, or >5 times), (**b**) Percentage of how often one of the top five used drugs was reported in studies compared to other drugs.

**Figure 5 pharmaceutics-13-00389-f005:**
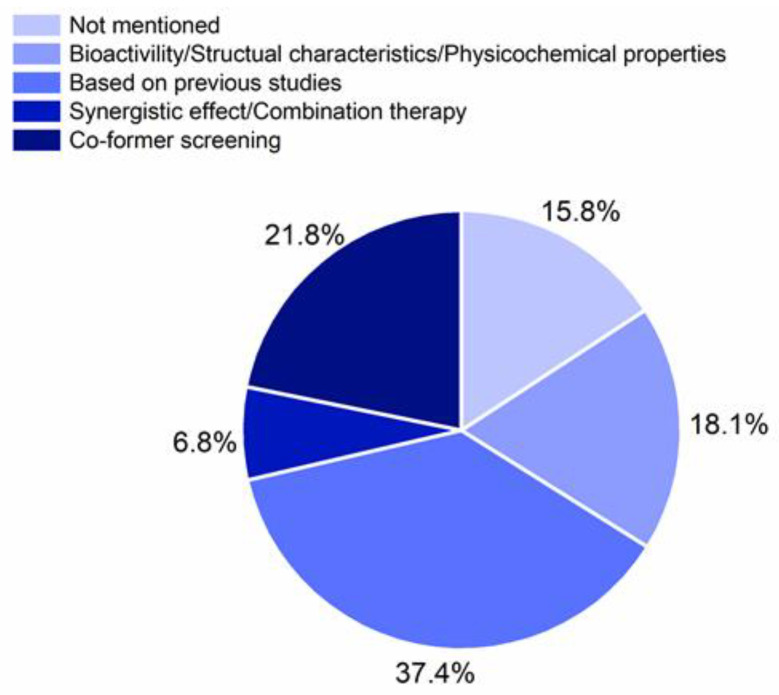
Percentage of stated rationales for co-former selection in the investigated CAMS.

**Figure 6 pharmaceutics-13-00389-f006:**
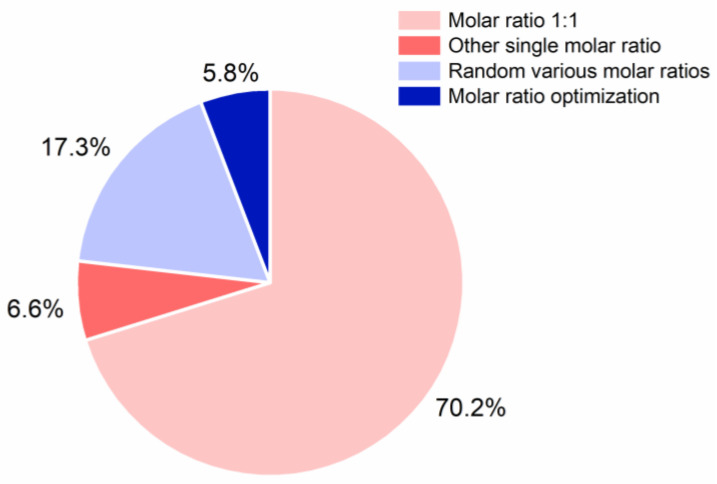
Percentage of studies using certain molar ratios, various molar ratios, and molar ratio optimization in the reported CAMS.

**Figure 7 pharmaceutics-13-00389-f007:**
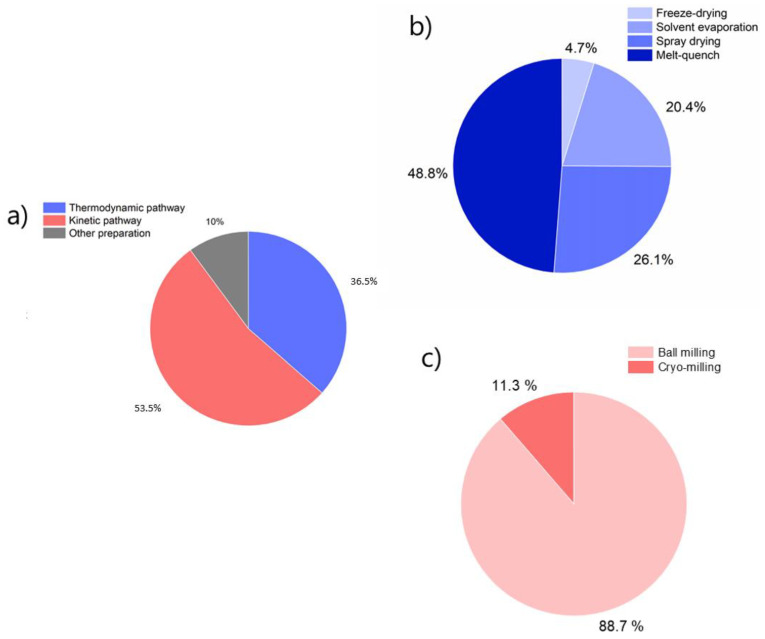
Percentage of the different preparation methods used to obtain CAMS in the reviewed studies, (**a**) Using the thermodynamic or kinetic pathway or other preparation methods, (**b**) Using different preparation methods of the thermodynamic pathways, and (**c**) Using different preparation methods of the kinetic pathways.

**Figure 8 pharmaceutics-13-00389-f008:**
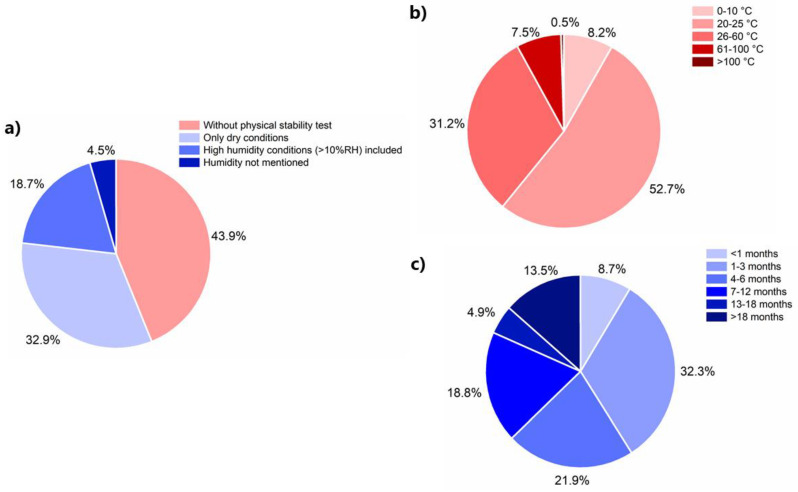
(**a**) Percentage of studies including physical stability tests or not and under which humidity conditions, (**b**) percentage of studies that included physical stability tests with respect to the storage temperature, and (**c**) percentage of studies that included physical stability tests with respect to the length of the storage period.

**Figure 9 pharmaceutics-13-00389-f009:**
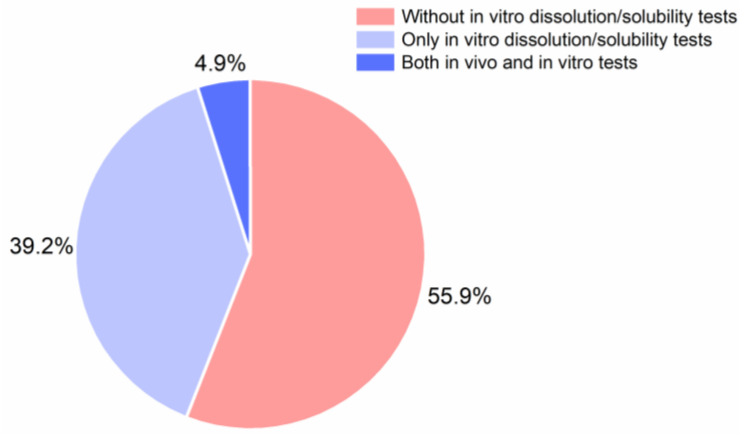
Percentage of studies with and without in vitro dissolution/solubility tests and with in vitro dissolution and in vivo tests.

**Figure 10 pharmaceutics-13-00389-f010:**
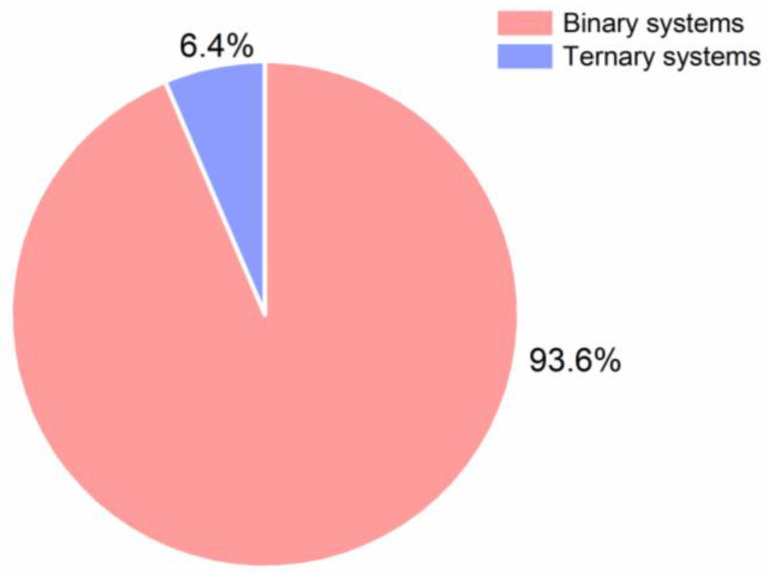
Percentage of binary CAMS and ternary CAMS in the reviewed literature.

**Table 3 pharmaceutics-13-00389-t003:** CAMS reported between a drug and an acidic amino acid as a co-former.

Drug	Amino Acid	Co- Amorphous (Y/N)	Preparation Method	Molar Ratio	Reference
Carvedilol	Aspartic acid	Y	Spray drying	1:1 [103], 2:1 to 1:4 [72]	[72,103]
Carvedilol	Aspartic acid	Y	Spray drying	With HPMC; 1:1, 1:1.5, 1:2	[104]
Carvedilol	Glutamic acid	Y	Spray drying	1:1 [103], 2:1 to 1:4 [72]	[72,103]

**Table 4 pharmaceutics-13-00389-t004:** CAMS reported between a drug and a polar amino acid as a co-former.

Drug	Amino Acid	Co- Amorphous (Y/N)	Preparation Method	Molar Ratio	Reference
Glibenclamide	Serine	Y	Cryo- milling	1:1	[28,115,116]
Glibenclamide	Threonine	Y	Cryo- milling	1:1	[28,115]

**Table 5 pharmaceutics-13-00389-t005:** CAMS reported between a basic drug and a basic amino acid as a co-former, as well as no formation of CAMS reported between an acidic drug and a basic amino acid as a co-former. LAG: Liquid-assisted grinding. * CAMS between glimepiride and arginine was formed using the supercritical antisolvent technique with supercritical carbon dioxide.

Drug	Amino Acid	Co- Amorphous (Y/N)	Preparation Method	Molar Ratio	Reference
Ibrutinib	Arginine	N	LAG	1:1	[36]
Cimetidine	Arginine	Y	Ball milling	1:1	[117]
Glimepiride	Arginine	N	Ball milling, Melt-quench, Solvent evaporation	1:1	[87] *
Mebendazole	Arginine	Y, but two-phase system	Ball milling	1:1	[117]
Ibrutinib	Histidine	N	LAG	1:2	[36]
Indomethacin	Histidine	N	Ball milling	1:1	[103]
Glibenclamide	Lysine	N	Cryo-milling	1:1	[28]
Indomethacin	Lysine	N	Ball milling	1:1	[103]

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
