# Peer review of "Co-Amorphous Drug Formulations in Numbers: Recent Advances in Co-Amorphous Drug Formulations with Focus on Co-Formability, Molar Ratio, Preparation Methods, Physical Stability, In Vitro and In Vivo Performance, and New Formulation Strategies"

_pharmaceutics, 2021, doi:10.3390/pharmaceutics13030389_

Round 1
Reviewer 1 Report
This is very useful paper, which gives an overview of co-amorphous systems. In this critical review authors include all relevant references regarding application of co-amorphous systems for delivery of poorly soluble drugs. The paper is well written and can be accepted in the present form.
Reviewer 2 Report
This is a comprehensive review which covers an interesting and emerging technology, i.e., coamorphization, in modulating pharmaceutical properties for improved drug delivery. The review is organized and well written, embracing all key topics related to coamorphous systems in pharmaceutics. The authors also nicely to summarize different examples and analyze their pharmaceutical properties (physical stability, in vitro and in vivo performance). Overall I recommend the work to be published in Pharmaceutics and I just have a few points for authors’ consideration for minor amendment:
Specific comments:
- Lines 65-67: The authors define the CAMS as the amorphous form of “a” drug by one or more low-molecular weight excipients. In this case, drug-drug coamorphous will be excluded. I suggest the definition should be slightly revised to include such case. Otherwise, this may not be consistent to lines 126 (which includes drug-drug CAMS). Also, some examples listed in Table 7, e.g., sacubitril-valsartan and budesonide-theophylline systems, may not fall within this definition.
- Section 6.1.2: I think the experimental Tg is a kinetic parameter which is also depending on the scanning rate. I think the authors may also include this point when they talk about the deviations although the reviewer agree the type of interaction plays a pivotal role.
- Line 635: The recrystallization here refers to “recrystallization of coamorphous system to cocrystals” and/or “recrystallization of coamorphous to individual crystalline component” ? Just out of my curiosity, is recrystallization from coamorphous to drug polymorph (eps. the unstable form) happen frequently from literature reported systems (based on Ostwald’s rule of stage)?
- Lines 655-659: Could the authors state what could be the reasons the “superior in vitro dissolution performance of a CAMS compared to the pure amorphous form and crystalline state of the drug is not guaranteed to translate into a better bioavailability, or even in vivo dissolution” in the review based on the references/their prior knowledge in the review ?
